# Identify Ambiguous Tasks Combining Crowdsourced Labels by Weighting Areas Under the Margin

**Tanguy Lefort**                                                       *tanguy.lefort@umontpellier.fr*
*IMAG, Univ. Montpellier, CNRS, LIRMM, INRIA*

**Benjamin Charlier**                                            *benjamin.charlier@umontpellier.fr*
*IMAG, Univ. Montpellier, CNRS*

**Alexis Joly**                                                                     *alexis.joly@inria.fr*
*LIRMM, INRIA*

**Joseph Salmon**                                                    *joseph.salmon@umontpellier.fr*
*IMAG, Univ. Montpellier, CNRS, Institut Universitaire de France (IUF)*

**Reviewed on OpenReview:** *https://openreview.net/forum?id=raD846nj2q*

## Abstract

In supervised learning — for instance in image classification — modern massive datasets are commonly labeled by a crowd of workers. The obtained labels in this crowdsourcing setting are then aggregated for training, generally leveraging a per-worker trust score. Yet, such workers oriented approaches discard the tasks' ambiguity. Ambiguous tasks might fool expert workers, which often impacts the learning step. In standard supervised learning settings – with one label per task – the Area Under the Margin (AUM) was tailored to identify mislabeled data. We adapt the AUM to identify ambiguous tasks in crowdsourced learning scenarios, introducing the Weighted Areas Under the Margin (WAUM). The WAUM is an average of AUMs weighted according to task-dependent scores. We show that the WAUM can help discarding ambiguous tasks from the training set, leading to better generalization performance. We report improvements over existing strategies for learning with a crowd, both on simulated settings, and on real datasets such as `CIFAR-10H` (a crowdsourced dataset with a high number of answered labels), `LabelMe` and `Music` (two datasets with few answered votes).

## 1  Introduction

Crowdsourcing labels for supervised learning has become quite common in the last two decades, notably for image classification datasets. Using a crowd of workers is fast, simple (see Figure 1) and less expensive than using experts. Furthermore, aggregating crowdsourced labels instead of working directly with a single one enables modeling the sources of possible ambiguities and directly taking them into account at training (Aitchison, 2021). With deep neural networks nowadays common in many applications, both the architectures and data quality have a direct impact on the model performance (Müller et al., 2019; Northcutt et al., 2021a) and on calibration (Guo et al., 2017). Yet, depending on the crowd and platform's control mechanisms, the quality of the labels might be low, with possibly many mislabeled instances (Müller & Markert, 2019), hence, degrading generalization (Snow et al., 2008).

Popular label aggregation schemes take into account the uncertainty related to workers' abilities: for example by estimating confusions between classes, or using a latent variable representing each worker trust (Dawid & Skene, 1979; Kim & Ghahramani, 2012; Sinha et al., 2018; Camilleri & Williams, 2019). This leads to scoring workers without taking into account the inherent difficulty of the tasks at stake. Inspired by the Item

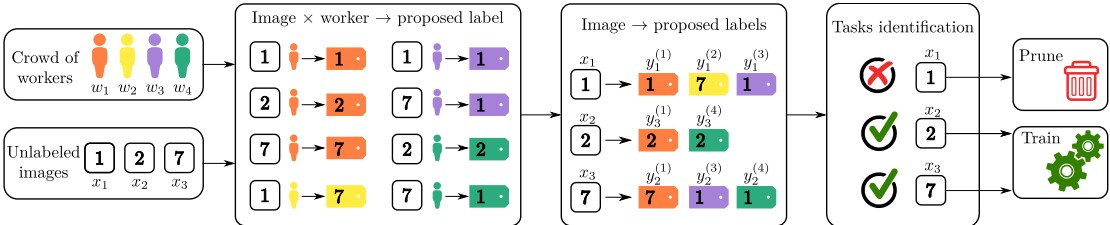

Figure 1: Learning with crowdsourcing labels: from label collection with a crowd to training on a pruned dataset. High ambiguity from either crowd workers or tasks intrinsic difficulty can lead to mislabeled data and harm generalization performance. To illustrate our notation, here the set of tasks annotated by worker $w_3$ is $\mathcal{T}(w_3) = \{1, 3\}$ while the set of workers annotating task $x_3$ is $\mathcal{A}(x_3) = \{1, 3, 4\}$.

Response Theory (IRT) introduced in (Birnbaum, 1968), the authors of (Whitehill et al., 2009) have combined both the task difficulty and the worker's ability in a feature-blind fashion for label aggregation. Other feature-blind aggregation strategies exist using (rank-one) matrix completion techniques (Ma & Olshevsky, 2020; Ma et al., 2020) or pairwise co-occurrences (Ibrahim et al., 2019). Both rely on the work by Dawid & Skene (1979) and take into account worker abilities but neglect the task difficulty. All the feature-blind strategies only leverage the labels but discard the associated features to evaluate workers performance. For instance, GLAD (Whitehill et al., 2009) estimates a task difficulty without the actual task: its estimation only relies on the collected labels and not on the tasks themselves (in image-classification settings, this means the images are not considered for evaluating the task difficulty). Neglecting such task difficulty might become critical when the number of labels collected per task is small.

In this work, we aim at identifying ambiguous tasks from their associated features, hence discarding hurtful tasks (such as the ones illustrated on Figure 2b and Figure 2c). Recent works on data-cleaning in supervised learning (Han et al., 2019; Pleiss et al., 2020; Northcutt et al., 2021b) have shown that some images might be too corrupted or too ambiguous to be labeled by humans. Hence, one should not consider these tasks for label aggregation or learning since they might reduce generalization power; see for instance (Pleiss et al., 2020). Throughout this work, we consider the ambiguity of a task with the informal definition proposed by Angelova (2004) that fit standard learning frameworks: *"Difficult examples are those which obstruct the learning process or mislead the learning algorithm or those which are impossible to reconcile with the rest of the examples"*. This definition links back to with how Pleiss et al. (2020) detect corrupted samples using the area under the margin (AUM) during the training steps of a machine learning classifier. However, it is important to notice that, in this context, the task ambiguity is inherent to the classifier architecture, and thus might not exactly overlap with human-level difficulty.

In this work, we combine task difficulty scores with worker abilities scores, but we measure the task difficulty by incorporating feature information. We thus introduce the Weighted Area Under the Margin (WAUM), a generalization to the crowdsourcing setting of the Area Under the Margin (AUM) by (Pleiss et al., 2020). The AUM is a confidence indicator in an assigned label defined for each training task. It is computed as an average of margins over scores obtained along the learning steps. The AUM reflects how a learning procedure struggles to classify a task to an assigned label[1]. The AUM is well suited when training a neural network (where the steps are training epochs) or other iterative methods. For instance, it has led to better network calibration (Park & Caragea, 2022) using MixUp strategy (Zhang et al., 2018), *i.e.,* mixing tasks identified as simple and difficult by the AUM. The WAUM, our extension of the AUM, aims at identifying ambiguous data points in crowdsourced datasets, so one can prune ambiguous tasks that degrade the generalization. It is a weighted average of workers AUM, where the weights reflect trust scores based on task difficulty and workers' ability.

---

[1]See the Linear SVC in Figure 4 to visualize how the AUM is connected to the classical margin from the kernel literature.

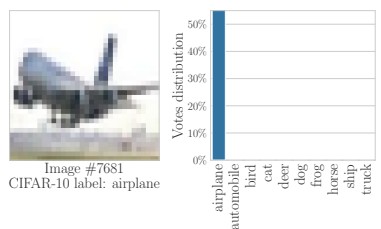

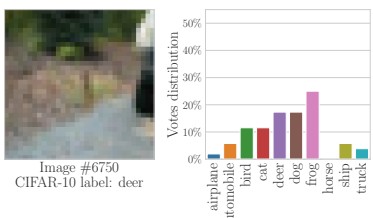

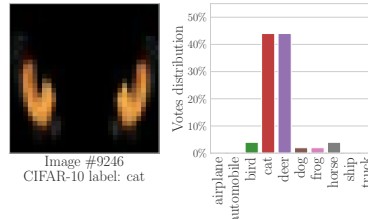

(a) Label `airplane` is easy to identify (unanimity among workers).

(b) Label `deer` is meaningless here, and workers are confused with all other labels.

(c) Label `cat` often confused with horns of a wild `deer`

Figure 2: Three images from `CIFAR-10H` dataset (Peterson et al., 2019), with the empirical distribution of workers' labels (soft labels): the `airplane` image (a) is easy, while the landscape (b) is ambiguous due to the image's poor quality. The last image (c) looks like a black cat face often perceived as the horns of a `deer`.

## 2 Related Work

Inferring a learning consensus from a crowd is a challenging task. In this work, we do not consider methods with prior knowledge on the workers, since most platforms do not provide this information[2]. Likewise, we do not rely on ground-truth knowledge for any tasks. Hence, trapping-set or control-items-based algorithms like ELICE or CLUBS (Khattak, 2017) do not match our framework. Some algorithms rely on self-reported confidence: they directly ask workers their answering confidence and integrate it into the model (Albert et al., 2012; Oyama et al., 2013; Hoang et al., 2021). We discard such cases for several reasons. First, self-reported confidence might not be beneficial without a reject option (Li & Varshney, 2017). Second, workers have a tendency to be under or overconfident, raising questions on how to present self-evaluation and assessing own scores (Draws et al., 2021).

To reach a consensus in the labeling process, the most common aggregation step is majority voting (MV), where one selects the label most often answered. MV does not infer any trust score on workers and does not leverage workers' abilities. MV is also very sensitive to under-performing workers (Gao & Zhou, 2013; Zhou et al., 2015), to biased workers (Kamar et al., 2015), to spammers (Raykar & Yu, 2011), or lack of experts for hard tasks (James, 1998; Gao & Zhou, 2013; Germain et al., 2015). Closely related to MV, naive soft (NS) labeling goes beyond *hard labels* (also referred to as *one-hot labels*) by computing the frequency of answers per label, yield a distribution over labels, often referred to as *soft-labels*. In practice, training a neural network with soft labels improves calibration (Guo et al., 2017) w.r.t. using hard labels. However, both MV and NS are sensitive to spammers (*e.g.,* workers answer *all* tasks randomly) or workers' biases (*e.g.,* workers who answer *some* tasks randomly). Hence, the noise induced by workers' labeling might not be representative of the actual task difficulty (Jamison & Gurevych, 2015).

Another class of methods leverages latent variables, defining a probabilistic model on workers' responses. The most popular one, proposed by (Dawid & Skene, 1979) (DS), estimates a single confusion matrix per worker, as a measure of workers' expertise. The underlying model assumes that a worker answers according to a multinomial distribution, yielding a joint estimation procedure of the confusion matrices and the soft labels through Expectation-Maximization (EM). Variants of the DS algorithm include accelerated (Sinha et al., 2018), sparse (Servajean et al., 2017), and clustered versions (Imamura et al., 2018) among others. DS strategy can also be used to create weights for a weighted MV (WMV) strategy. Indeed, each worker weight can be considered as how accurate they are at labeling the correct class, *i.e.* the sum of the diagonal of their confusion matrix.

Since DS only models workers' abilities, (Whitehill et al., 2009) have introduced the Generative model of Labels, Abilities, and Difficulties (GLAD) to exploit task difficulties to improve confusion estimation. While DS estimates a matrix of pairwise label confusion per worker, GLAD considers also an EM procedure to estimate one ability score per worker, and one difficulty score per task. It is inspired by the IRT (Birnbaum,

---

[2]For instance, by default Amazon Mechanical Turk `https://www.mturk.com/` does not provide it.

1968), modeling the workers' probability to answer the true label with a logistic transform of the product of these scores. Following IRT, the difficulty is inferred as a latent variable given the answers: as for DS, the underlying tasks are discarded. Other methods based on probabilistic models or objective optimization have been developed. Platanios et al. (2014) defines a setting to estimate the accuracy of a set of classifiers (workers) with unlabeled data based on their agreement/error rates. This could later be used as a set of worker weights in a different weighted majority vote, in a setting with dependent workers. Their agreement rate (AR) approach is based on binary classifications output, and the adaptation to multiple classes needs to be split into binary problems to optimize. Each of those optimization problems depends exponentially on the number of workers and thus is limited for our crowdsourced datasets as is. In Platanios et al. (2017), they later proposed a new probabilistic-logic approach for the same accuracy inference problem. They infer the output based on classifier (workers) outputs and a set of logical constraints. Multiclass classification fits this setting using mutual exclusion class constraints. They achieve competitive performance against probabilistic strategies and could in practice be combined with the pruning method presented in this paper.

Finally, following deep learning progresses, end-to-end strategies have emerged that do not produce aggregated labels but allow to train classifiers from crowdsourced labels. Rodrigues & Pereira (2018) introduced CrowdLayer adding a new layer inside the network mimicking confusion matrices per worker. Later, Chu et al. (2021) have generalized this setting with CoNAL, adding an element encoding global confusion.

Here, we propose the WAUM to combine the information from a confusion matrix per worker and a measure of relative difficulty between tasks. It refines the judging system and identifies data points harming generalization that should be pruned. Data pruning has been shown to improve generalization by removing mislabeled data (Angelova et al., 2005; Pleiss et al., 2020), possibly dynamically along the learning phase (Raju et al., 2021) or by defining a forgetfulness score (Paul et al., 2021). Recent work on data pruning in crowdsourcing (Xing et al., 2023) separates label errors (the tasks are useful for learning as their ambiguity is relevant) from label noise (tasks are too ambiguous for learning) in binary classification setting. They also consider the model scores for each task and the given label to estimate the probability for a given label to be the true label. And from these estimates, they rank the tasks and prune the most unreliable ones. In this paper, however, we only consider multiclass classification and can not consider as-is this concurrent strategy. Sorscher et al. (2022) have highlighted that data pruning strategies are highly impacted by the labeling in supervised settings and we confirm its relevance to the crowdsourcing framework. It is also a flexible tool that can be combined with most existing methods, using the pruning as a preliminary step.

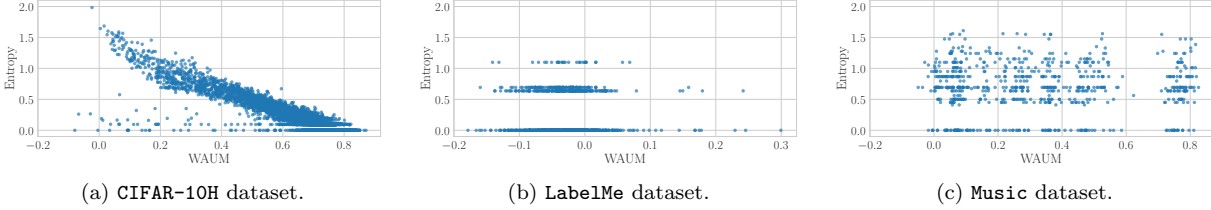

| (a) CIFAR-10H dataset. | (b) LabelMe dataset. | (c) Music dataset. |

Figure 3: Entropy of votes vs. WAUM for CIFAR-10H, LabelMe, and Music, each point representing a task/image. When large amounts of votes per task are available, WAUM and entropy ranking coincide well, as in (a). Yet, when votes are scarce, as in (b) and (c), entropy becomes irrelevant while our introduced WAUM remains useful. Indeed, tasks with few votes can benefit from feedback obtained for a similar one. And for the LabelMe dataset in particular, there are only up to three votes available per task, thus only four different values of the entropy possible, making it irrelevant in such cases for modeling task difficulty.

## 3 Weighted Area Under the Margin

### 3.1 Definitions and Notation

**General notation.** We consider classical multi-class learning notation, with input in $\mathcal{X}$ and labels in $[K] := \{1, \ldots, K\}$. The set of tasks is written as $\mathcal{X}_{\text{train}} = \{x_1, \ldots, x_{n_{\text{task}}}\}$, and we assume

$\{(x_1, y_1^\star), \ldots, (x_{n_{\texttt{task}}}, y_{n_{\texttt{task}}}^\star)\}$ are $n_{\texttt{task}}$ *i.i.d* tasks and labels, with underlying distribution denoted by $\mathbb{P}$. The true labels $(y_i^\star)_{i \in [n_{\texttt{task}}]}$ are unobserved but crowdsourced labels are provided by $n_{\texttt{worker}}$ workers $(w_j)_{j \in [n_{\texttt{worker}}]}$. We write $\mathcal{A}(x_i) = \{j \in [n_{\texttt{worker}}] : \text{worker } w_j \text{ labeled task } x_i\}$ the **annotators set** of a task $x_i$ and $\mathcal{T}(w_j) = \{i \in [n_{\texttt{task}}] : \text{worker } w_j \text{ answered task } x_i\}$ the **tasks set** for a worker $w_j$. For a task $x_i$ and each $j \in \mathcal{A}(x_i)$, we denote $y_i^{(j)} \in [K]$ the label answered by worker $w_j$. Given an aggregation strategy **agg** (such as MV, DS or GLAD), we call estimated soft label $\hat{y}_i^{\texttt{agg}}$ the obtained label. Note that for MV, the aggregated label $\hat{y}_i^{\text{MV}} \in [K]$ and for other strategies, $\hat{y}_i^{\texttt{agg}}$ lies in the standard simplex $\Delta_{K-1} = \{p \in \mathbb{R}^K, \sum_{k=1}^{K} p_k = 1, p_k \geq 0\}$. For any set $\mathcal{S}$, we write $|\mathcal{S}|$ for its cardinality. Examples of annotators set and tasks set are provided in Figure 1. The training set has task-wise and worker-wise formulations:

$$\mathcal{D}_{\texttt{train}} = \bigcup_{i=1}^{n_{\texttt{task}}} \left\{ \left(x_i, \left(y_i^{(j)}\right)\right) \text{ for } j \in \mathcal{A}(x_i) \right\} = \bigcup_{j=1}^{n_{\texttt{worker}}} \underbrace{\left\{ \left(x_i, \left(y_i^{(j)}\right)\right) \text{ for } i \in \mathcal{T}(w_j) \right\}}_{\mathcal{D}_{\texttt{train}}^{(j)}} . \quad (1)$$

**DS model notation.** The Dawid and Skene (DS) model (Dawid & Skene, 1979) aggregates answers and evaluates the workers' confusion matrix to observe where their expertise lies. The confusion matrix of worker $w_j$ is denoted by $\pi^{(j)} \in \mathbb{R}^{K \times K}$ and reflects individual error-rates between pairs of labels: $\pi_{\ell,k}^{(j)} = \mathbb{P}(y_i^{(j)} = k | y_i^\star = \ell)$ represents the probability that worker $w_j$ gives label $k$ to a task whose true label is $\ell$. The model assumes that the probability for a task $x_i$ to have true label $y_i^\star = \ell$ follows a multinomial distribution with probabilities $\pi_{\ell,\cdot}^{(j)}$ for each worker, independently of $\mathcal{X}_{\texttt{train}}$ (feature-blind). In practice, DS estimates are obtained thanks to the EM algorithm to output estimated confusion matrices $(\pi^{(j)})_{j \in [n_{worker}]}$. The full likelihood is given in Equation (8), Appendix A.3. Once DS confusion matrices are estimated, it is possible to use the diagonal terms as weights in a majority voting strategy. We denote this Weighted DS vote by WDS, and give more details in Appendix A.4. Essentially, the WDS strategy produces soft labels as NS, and also takes into account the estimated worker ability to recognize a task whose true label is indeed the voted one.

### 3.2 Ambiguous tasks identification with the AUM

To identify labeling errors and evaluate task difficulties, Pleiss et al. (2020) have introduced the AUM in the standard learning setting (*i.e.*, when $|\mathcal{A}(x_i)| = 1$ for all $i \in [n_{\texttt{task}}]$). Given a training task and a label $(x, y)$, let $z^{(t)}(x) \in \mathbb{R}^K$ be the logit score vector at epoch $t \leq T$ when learning a neural network (where $T$ is the number of training epochs). We use the notation $z_{[1]}^{(t)}(x) \geq \cdots \geq z_{[K]}^{(t)}(x)$ for sorting $(z_1^{(t)}(x), \ldots, z_K^{(t)}(x))$ in non-increasing order. Let us denote $\sigma^{(t)}(x) := \sigma(z^{(t)}(x))$ the softmax output of the scores at epoch $t$. Sorting the probabilities in decreasing order such that $\sigma_{[1]}^{(t)}(x) \geq \cdots \geq \sigma_{[K]}^{(t)}(x)$, the AUM reads:

$$\text{AUM}\,(x, y; \mathcal{D}_{\texttt{train}}) = \frac{1}{T} \sum_{t=1}^{T} [\sigma_y^{(t)}(x) - \sigma_{[2]}^{(t)}(x)] . \quad (2)$$

We write $\text{AUM}\,(x, y)$ instead of $\text{AUM}\,(x, y; \mathcal{D}_{\texttt{train}})$ when the training set is clear from the context. Pleiss et al. (2020) use an average of margins over logit scores, while we rather consider the average of margin after a softmax step in Equation (2), to temper scaling issues, as advocated by Ju et al. (2018) in ensemble learning. Moreover, we consider the margin introduced by Yang & Koyejo (2020) since the corresponding hinge loss has better theoretical properties than the one used in the original AUM, especially in top-$k$ settings[3] (Lapin et al., 2016; Yang & Koyejo, 2020; Garcin et al., 2022). However, one could easily consider the original margin with few differences in practice for top-1 classification (see Appendix G).

During the training phase, the AUM keeps track of the difference between the score assigned to the proposed label and the score assigned to the second-largest one. It has been introduced to detect mislabeled observations in a dataset: The higher the AUM, the more likely the network confirms the given label. And, the lower the AUM, the harder it is for the network to differentiate the given label from (at least) another class.

---

[3]For top-$k$, consider $\sigma_{[k+1]}^{(t)}(x)$ instead of $\sigma_{[2]}^{(t)}(x)$ in equation 2.

Finally, note that the AUM computation depends on the chosen neural network and on its initialization: pre-trained architectures could be used, yet any present bias would transfer to the AUM computation.

To generalize the AUM from Equation (2) to the crowdsourcing setting, a difficulty lies in the term $\sigma_y^{(t)}(x)$ as, in this context, the label $y$ is unknown, as one observes several labels per task. A naive adaptation of the AUM would be to use the majority voting strategy in order to recover a hard label to be used in Equation (2). We denote such a strategy by AUMC (AUM for Crowdsourced data). More formally, this writes as:

$$\text{AUMC}\left(x_i, \left\{y_i^{(j)}\right\}_{j \in \mathcal{A}(x_i)}; \mathcal{D}_{\texttt{train}}\right) = \frac{1}{T}\sum_{t=1}^{T}\left[\sigma_{\hat{y}_i^{\text{MV}}}^{(t)}(x_i) - \sigma_{[2]}^{(t)}(x_i)\right] \ . \tag{3}$$

This naive approach can be refined by taking into account the whole distribution of labels, and not simply its mode (with MV).

### 3.3 WAUM and data pruning

The AUM is defined in a standard supervised setting with (hard) labels. The naive adaptation AUMC defined at equation 3 does not take into account the fact that workers may have different abilities. We now adapt the AUM to crowdsourced frameworks to improve the identification of difficult tasks. Mirroring the AUM, train a classifier network on $\mathcal{D}_{\texttt{train}}$ for a number $T > 0$ of epochs. Using this classifier, we can compute the following. Let $s^{(j)}(x_i) \in [0, 1]$ be a trust factor in the answer of worker $w_j$ for task $x_i$. The WAUM is then defined as:

$$\text{WAUM}(x_i) = \frac{\displaystyle\sum_{j \in \mathcal{A}(x_i)} s^{(j)}(x_i)\text{AUM}\big(x_i, y_i^{(j)}\big)}{\displaystyle\sum_{j' \in \mathcal{A}(x_i)} s^{(j')}(x_i)} \ . \tag{4}$$

It is a weighted average of AUMs over each worker's answer with a per task weighting score $s^{(j)}(x_i)$ based on workers' abilities. This score considers the impact of the AUM for each answer since it is more informative if the AUM indicates uncertainty for an expert than for a non-expert.

The scores $s^{(j)}$ are obtained *à la* Servajean et al. (2017): each worker has an estimated confusion matrix $\hat{\pi}^{(j)} \in \mathbb{R}^{K \times K}$. Note that the vector $\text{diag}(\hat{\pi}^{(j)}) \in \mathbb{R}^K$ represents the probability for worker $w_j$ to answer correctly to each label. With a neural network classifier, we estimate the probability for the input $x_i \in \mathcal{X}_{\texttt{train}}$ to belong in each category by $\sigma^{(T)}(x_i)$, *i.e.*, the probability estimate at the last epoch. As a trust factor, we propose the inner product between the diagonal of the confusion matrix and the softmax vector:

$$s^{(j)}(x_i) = \big\langle \text{diag}(\hat{\pi}^{(j)}), \sigma^{(T)}(x_i) \big\rangle \in [0, 1] \ . \tag{5}$$

The scores control the weight of each worker in Equation (4). This choice of weight is inspired by the bilinear scoring system of GLAD (Whitehill et al., 2009), as detailed hereafter. The closer to one, the more we trust the worker for the given task. The score $s^{(j)}(x_i)$ can be seen as a multidimensional version of GLAD's trust score. Indeed, in GLAD, the trust score is modeled as the product $\alpha_j \beta_i$, with $\alpha_j \in \mathbb{R}$ (resp. $\beta_i \in (0, +\infty)$) representing worker ability (resp. task difficulty). In Equation (5), the diagonal of the confusion matrix $\hat{\pi}^{(j)}$ represents the worker's ability and the softmax the task difficulty.

**Dataset Pruning and hyperparameter tuning.** Our procedure (Algorithm 1) proceeds as follows. We initialize our method by estimating the confusion matrices for all workers. For each worker $w_j$, the AUM is computed for its labeled tasks, and so is its worker-dependent trust scores $s^{(j)}(x_i)$ with Equation (5) during the training phase of a classifier. The WAUM in Equation (4) is then computed for each task. The most ambiguous tasks, the ones whose WAUM are below a threshold, are then discarded, and the associated pruned dataset $\mathcal{D}_{\text{pruned}}$ is output. We consider for the pruning threshold a quantile of order $\alpha \in [0, 1]$ of the WAUM scores. The hyperparameter $\alpha$ (proportion of training data points pruned) is calibrated on a validation set, choosing $\alpha \in \{0.1, 0.05, 0.01\}$ has led to satisfactory results in all our experiments. In general, dataset annotated by humans have roughly between 1 and 5% of errors (Northcutt et al., 2021a) and the choice

---

**Algorithm 1** WAUM (Weighted Area Under the Margin).

---

**Input**: $\mathcal{D}_{\texttt{train}}$: tasks and crowdsourced labels, $\alpha \in [0,1]$: proportion of training points pruned, $T \in \mathbb{N}$: number of epochs, $\texttt{Est}$: Estimation procedure for the confusion matrices

**Output**: pruned dataset $\mathcal{D}_{\text{pruned}}$

1: Get confusion matrix $\{\hat{\pi}^{(j)}\}_{j \in [n_{\texttt{worker}}]}$ from $\texttt{Est}$

2: Train a classifier for $T$ epochs on $\left(x_i, y_i^{(j)}\right)_{i,j}$

3: **for** $j \in [n_{\text{worker}}]$ **do**

4:     Get $\text{AUM}(x_i, y_i^{(j)}; \mathcal{D}_{\texttt{train}})$ using Equation (2) for $i \in \mathcal{T}(w_j)$

5:     Get **trust scores** $s^{(j)}(x_i)$ using Equation (5) for $i \in \mathcal{T}(w_j)$

6: **end for**

7: **for** each task $x \in \mathcal{X}_{\texttt{train}}$ **do**

8:     Compute $\text{WAUM}(x)$ using Equation (4)

9: **end for**

10: Get $q_\alpha \left(\text{WAUM}(x_i)\right)_{i \in [n_{\text{task}}]}$, $\alpha$-**quantile threshold**

11: $\mathcal{D}_{\text{pruned}} = \left\{\left(x_i, \left(y_i^{(j)}\right)_{j \in \mathcal{A}(x_i)}\right) : \text{WAUM}(x_i) \geq q_\alpha, x_i \in \mathcal{X}_{\texttt{train}}\right\}$

---

of $\alpha$ should reflect that. Note that the same pruning procedure can be applied to AUMC for comparison. Both the AUMC and WAUM inherit the hyperparameter $T > 0$ from the original AUM. Following the recommendations from Pleiss et al. (2020), we need $T$ large enough for stability and $T$ not too big to avoid overfitting the data. In practice, a guideline given is to train until the first learning rate scheduler drop to only keep the beginning of the scores trajectories without finetuning. The main assumptions to identify ambiguous tasks is thus not to over-train the neural network in the WAUM (or AUMC) step, and be able to run a DS-like algorithm to recover the diagonal of the confusion matrix for Equation (5).

**Refined initialization: estimating confusion matrices.** By default, we rely on the $\texttt{Est}$=DS algorithm to get workers' confusion matrices, but other estimates are possible: DS might suffer from the curse of dimensionality when the number $K$ of classes is large ($K^2$ coefficients needed per worker).

**Training on the pruned dataset** Once a pruned dataset $\mathcal{D}_{\text{pruned}}$ has been obtained thanks to the WAUM, one can create soft labels through an aggregation step, and use them to train a classifier. Aggregated soft labels contain information regarding human uncertainty, and could often be less noisy than NS labels. They can help improve model calibration (Wen et al., 2021; Zhong et al., 2021), a property useful for interpretation (Jiang et al., 2012; Kumar et al., 2019). Concerning the classifier training, note that it can differ from the one used to compute the WAUM. We train a neural network whose architecture is adapted dataset per dataset and that can differ from the one used in Algorithm 1 (it is the case for instance for the $\texttt{LabelMe}$ dataset). For an aggregation technique $\texttt{agg}$, we write the full training method on the pruned dataset created from the WAUM: $\texttt{agg} + \text{WAUM}$ and instantiate several choices in Section 4. For comparison, we write $\texttt{agg} + \text{AUMC}$ the training method on the pruned dataset created from the AUMC.

## 4 Experiments

Our first experiments focus on multi-class classification datasets with a large number of votes per task. We consider first a simulated dataset to investigate the WAUM and the pruning hyperparameter $\alpha$. Then, with the real $\texttt{CIFAR-10H}$ dataset from Peterson et al. (2019) we compare label aggregation-based procedures with and without pruning using the AUMC or the WAUM. Finally, we run our experiments on the $\texttt{LabelMe}$ dataset from Rodrigues & Pereira (2018) and $\texttt{Music}$ dataset from Rodrigues et al. (2014), both real crowdsourced datasets with few labels answered per task. For each aggregation scheme considered, we train a neural network on the soft labels (or hard labels for MV) obtained after the aggregation step. We compare our WAUM scheme with several other strategies like GLAD (feature-blind) or CoNAL (feature-aware) with and without pruning from the AUMC identification step. For CoNAL, two regularization levels are considered:

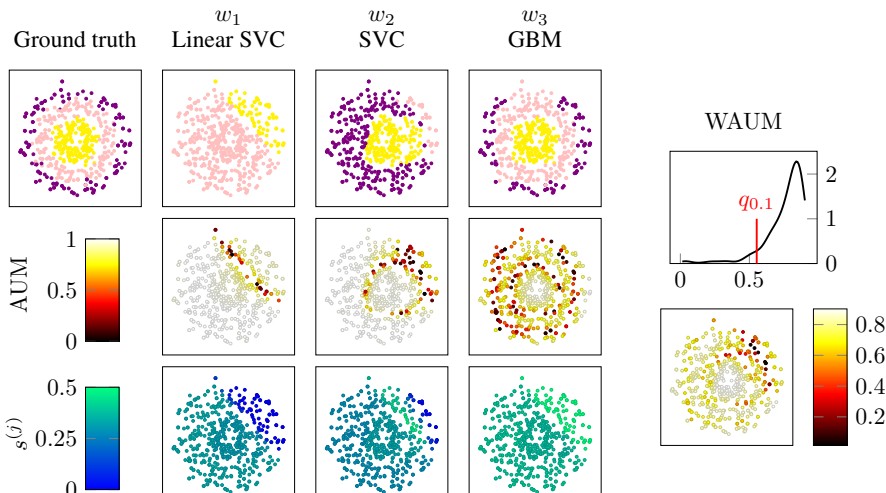

Figure 4: `three_circles`: one realization of simulated workers $w_1, w_2, w_3$, with their AUM, normalized trust scores $s^{(j)}$ (left) and WAUM distributions (right) for $\alpha = 0.1$. Worker $w_1$ has less impact into the final WAUM in the disagreement area. Note also that for worker $w_1$ (LinearSVC), the region with low AUM values recovers the usual classifier's margin around the decision boundary.

$\lambda = 0$ and $\lambda = 10^{-4}$ ($\lambda$ controls the distance between the global and the individual confusion matrices). More simulations and overview of the methods compared are available in Appendix D.1.

**Metrics investigated**   After training, we report two performance metrics on a test set $\mathcal{D}_{\texttt{test}}$: top-1 accuracy and expected calibration error (ECE) (with $M = 15$ bins as in Guo et al. (2017)). The ECE measures the discrepancy between the predicted probabilities and the probabilities of the underlying distribution. For ease of reporting results, we display the score $1 - \text{ECE}$ (hence, the higher the better, and the closer to 1, the better the calibration); see Appendix C for more details. Reported errors represent standard deviations over the repeated experiments (10 repetitions on simulated datasets and 3 for real datasets).

**Implementation details**   For simulations, the training is performed with a three dense layers' artificial neural network (a MLP with three layers) $(30, 20, 20)$ with batch size set to 64. Workers are simulated with `scikit-learn` (Pedregosa et al., 2011) classical classifiers. For `CIFAR-10H` the Resnet-18 (He et al., 2016) architecture is chosen with batch size set to 64. We minimize the cross-entropy loss, and use when available a validation step to avoid overfitting. For optimization, we consider an `SGD` solver with 150 training epochs, an initial learning rate of 0.1, decreasing it by a factor 10 at epochs 50 and 100. The WAUM and AUMC are computed with the same parameters for $T = 50$ epochs. Other hyperparameters for `Pytorch`'s (Paszke et al., 2019) `SGD` are `momentum=0.9` and `weight_decay=5e-4`. For the `LabelMe` and `Music` datasets, we use the Adam optimizer with learning rate set to 0.005 and default hyperparameters. On these two datasets, the WAUM and AUMC are computed using a more classical Resnet-50 for $T = 500$ epochs and the same optimization settings. The architecture used for train and test steps is a pretrained VGG-16 combined with two dense layers as described in Rodrigues & Pereira (2018) to reproduce original experiments on the `LabelMe` dataset. This architecture differs from the one used to recover the pruned set. Indeed, contrary to the modified VGG-16, the Resnet-50 could be fully pre-trained. The general stability of pre-trained Resnets, thanks to the residuals connections, allows us to compute the WAUM and AUMC with way fewer epochs (each being also with a lower computational cost) compared to VGGs (He et al., 2016). As there are few tasks, we use data augmentation with random flipping, shearing and dropout (0.5) for 1000 epochs. Experiments were executed with Nvidia RTX 2080 and Quadro T2000 GPUs. Appendix B presents more details on the code used with the `peerannot` library. Source codes are available at `https://github.com/peerannot/peerannot`. Evaluated strategies are at `https://github.com/peerannot/peerannot/tree/main/peerannot/models` sorted according to whether they are aggregation-based, learning-based or only for identification. The WAUM and AUMC sources are available in the `identification` module.

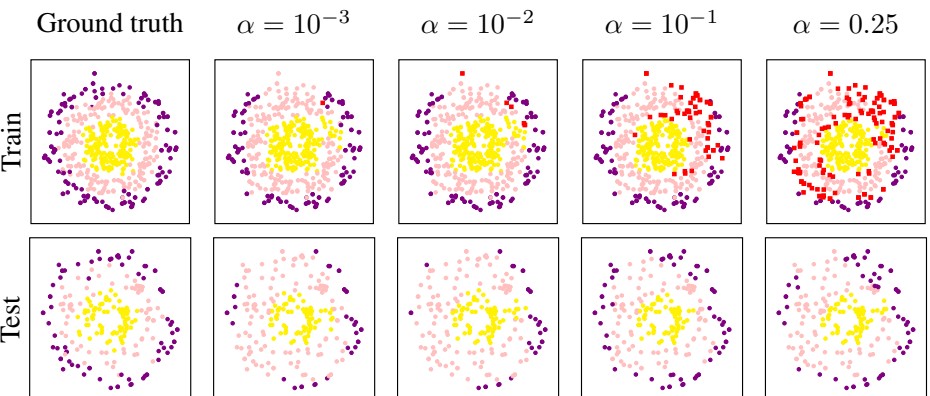

Figure 5: Influence of $\alpha$ on the pruning step. Red dots indicate data points pruned from the training set, at level $q_\alpha$ in the WAUM (see line 10 in Algorithm 1). We consider ($\alpha \in \{10^{-3}, 10^{-2}, 10^{-1}, 0.25\}$). The neural network used for predictions is three dense layers' $(30, 20, 20)$, as for other simulated experiments. Training labels are from the WDS + WAUM strategy with performance reported in Section 4.1. The more we prune data, the worse the neural network can learn from the training dataset. However, removing the tasks with high disagreement noise helps to generalize.

## 4.1 Simulated multiclass dataset: `three_circles`.

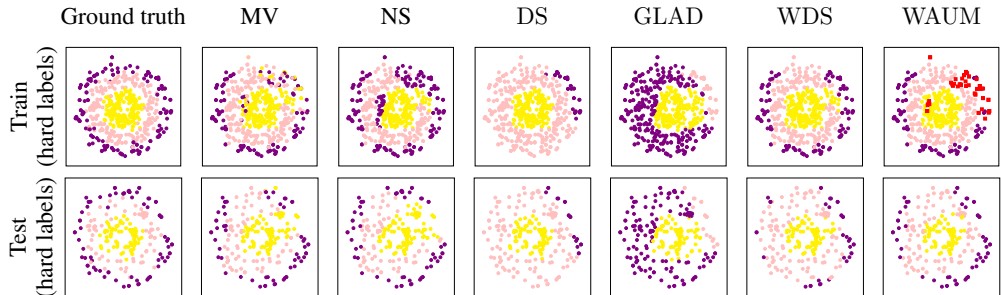

Figure 6: `three_circles`: One realization of Section 4.1 varying the aggregation strategy. Training labels are provided from Figure 4 and predictions on the test set are from three dense layers' artificial neural network $(30, 20, 20)$ trained on the aggregated soft labels. For ease of visualization, the color displayed for each task represents the most likely class. Red points are pruned from training by WAUM with threshold $\alpha = 0.1$. Here, we have $n_{\texttt{task}} = 525$. WAUM method as in Section 4.1 uses WDS labels.

We simulate three cloud points (to represent $K = 3$ classes) using `scikit-learn`'s function `two_circles`; see Figure 4. The $n_{\texttt{worker}} = 3$ workers are standard classifiers: $w_1$ is a linear Support Vector Machine Classifier (linear SVC), $w_2$ is an SVM with RBF kernel (SVC), and $w_3$ is a gradient boosted classifier (GBM). Data is split between train (70%) and test (30%) for a total of 750 points and each simulated worker votes for all tasks, *i.e.,* for all $x \in \mathcal{X}_{\texttt{train}}$, $|\mathcal{A}(x)| = n_{\texttt{worker}} = 3$, leading to $n_{\texttt{task}} = 525$ tasks (points).We do not use a validation set to calibrate the hyperparameter $\alpha$ as this experiment is mostly for a pedagogical purpose. The performance reported in Section 4.1 is averaged over 10 repetitions using the `peerannot` library.

A disagreement area is identified in the northeast area of the dataset (see Figure 4). Section 4.1 also shows that pruning too little data ($\alpha$ small) or too much ($\alpha$ large) can mitigate the performance. In Figure 5, we show the impact of the pruning hyperparameter $\alpha$. The closer $\alpha$ is to 1, the more training tasks are pruned from the training set (and the worse the performance).

| Strategy | $\text{Acc}_{\text{test}}$ | ECE |
|---|---|---|
| MV | $0.73 \pm 0.03$ | $\mathbf{0.13} \pm \mathbf{0.03}$ |
| NS | $0.70 \pm 0.02$ | $0.18 \pm 0.02$ |
| DS | $0.75 \pm 0.07$ | $0.22 \pm 0.08$ |
| GLAD | $0.58 \pm 0.02$ | $0.36 \pm 0.02$ |
| WDS | $0.81 \pm 0.04$ | $0.17 \pm 0.03$ |
| WDS + AUMC($\alpha = 10^{-1}$) | $0.81 \pm 0.02$ | $0.17 \pm 0.01$ |
| WDS + WAUM($\alpha = 10^{-2}$) | $0.80 \pm 0.04$ | $0.17 \pm 0.01$ |
| WDS + WAUM($\alpha = 10^{-1}$) | $\mathbf{0.83} \pm \mathbf{0.03}$ | $0.19 \pm 0.04$ |
| WDS + WAUM($\alpha = 0.25$) | $0.69 \pm 0.02$ | $0.19 \pm 0.02$ |

Table 1: `three_circles`: Aggregation and learning performance presented in Figure 6 ($n_{\text{task}} = 525$ tasks, $|\mathcal{A}(x)| = n_{\text{worker}} = 3$, 10 repetitions). Errors represented are standard deviations. Note that the best worker, $w_3$, reaches 0.84 on test accuracy. We vary $\alpha \in \{0.01, 0.1, 0.25\}$ to visualize the impact of pruning.

## 4.2 Real datasets

In this section, we investigate three popular crowdsourced datasets: `CIFAR-10H`, `LabelMe` and `Music`. The first one, `CIFAR-10H` (Peterson et al., 2019), is a curated dataset with many votes per task while `LabelMe` (Rodrigues & Pereira, 2018) and `Music` (Rodrigues et al., 2014) datasets are more challenging, having fewer labels per task. This low number of votes per task, especially for `LabelMe` can lead to erroneous MV label which then impact the quality of the AUMC. In this context, the label distribution's entropy is also a poor choice to identify hard tasks as can be seen in Figure 3. Indeed, with up to three labels, the entropy can only take four different values and thus is no help in ranking the difficulty of 1000 tasks.

To prune only a few tasks, we have $\alpha = 1\%$ for `CIFAR-10H` and `LabelMe` datasets. For the `Music` dataset, $\alpha = 5\%$ leads to better generalization performance; considering the dataset size and complexity, picking $\alpha = 0.1$ would lead to worse performance. This value for the hyperparameter $\alpha$ has been calibrated as described in Section 3.3 using the accuracy on the available validation set for $\alpha \in \{0.01, 0.05, 0.1\}$. Ablation studies by architecture are performed on `CIFAR-10H` and `LabelMe` datasets in Figure 12 to show consistent improvement in performance by using the WAUM to prune ambiguous data.

| Strategy | $\text{Acc}_{\text{test}}(\%)$ | $1 - \text{ECE}$ |
|---|---|---|
| MV | $69.53 \pm 0.84$ | $0.825 \pm 0.00$ |
| MV + AUMC | $71.12 \pm 1.12$ | $\mathbf{0.836} \pm \mathbf{0.01}$ |
| MV + WAUM | $\mathbf{72.34} \pm \mathbf{1.01}$ | $0.814 \pm 0.02$ |
| NS | $72.14 \pm 2.74$ | $\mathbf{0.868} \pm \mathbf{0.03}$ |
| NS + AUMC | $71.80 \pm 2.12$ | $0.838 \pm 0.00$ |
| NS + WAUM | $\mathbf{72.21} \pm \mathbf{1.82}$ | $0.829 \pm 0.00$ |
| DS | $70.26 \pm 0.93$ | $0.827 \pm 0.00$ |
| DS + AUMC | $70.43 \pm 1.10$ | $\mathbf{0.833} \pm \mathbf{0.02}$ |
| DS + WAUM | $\mathbf{72.71} \pm \mathbf{0.98}$ | $0.814 \pm 0.02$ |
| GLAD | $70.28 \pm 0.88$ | $\mathbf{0.838} \pm \mathbf{0.01}$ |
| GLAD + AUMC | $70.42 \pm 1.23$ | $0.830 \pm 0.01$ |
| GLAD + WAUM | $\mathbf{71.93} \pm \mathbf{1.12}$ | $0.812 \pm 0.02$ |
| WDS | $72.49 \pm 0.48$ | $\mathbf{0.868} \pm \mathbf{0.00}$ |
| WDS + AUMC | $72.47 \pm 0.45$ | $0.866 \pm 0.00$ |
| WDS + WAUM | $\mathbf{72.67} \pm \mathbf{0.59}$ | $\mathbf{0.868} \pm \mathbf{0.00}$ |

Table 2: `CIFAR-10H`: performance of a `ResNet-18` by label-aggregation crowdsourcing strategy ($\alpha = 0.01$). Errors represented are standard deviations.

**`CIFAR-10H` dataset.** The training part of `CIFAR-10H` consists of the 10000 tasks extracted from the test set of the classical `CIFAR-10` dataset (Krizhevsky & Hinton, 2009), and $K = 10$. A total of $n_{\text{worker}} = 2571$ workers participated on the Amazon Mechanical Turk platform, each labeling 200 images (20 from each original class), leading to approximately 50 answers per task. We have randomly extracted 500 tasks for a validation set (hence $n_{\text{train}} = 9500$). The test set of `CIFAR-10H` is comprised of the train set of `CIFAR-10` (see more details in Appendix D.2). This dataset is notoriously more curated (Aitchison, 2021) than a common

dataset in the field: most difficult tasks were identified and removed at the creation of the `CIFAR-10` dataset, resulting in few ambiguities (Krizhevsky & Hinton, 2009). Section 4.2 shows that in this simple setting, our data pruning strategy is still relevant, with the choice $\alpha = 0.01$. Images with worst WAUM for each class are presented in Figure 7.

Furthermore, the WAUM leads to better generalization performance than the vanilla DS model and the pruning with AUMC. Overall, we show that there is a gain in performance to obtain by using a pruning pre-processing step compared to training the classifier on the aggregated labels for the full training set. There is consistently an improvement on using the WAUM pruning – which weights the margins by worker and tasks – over the naive AUMC which does not use reweighing.

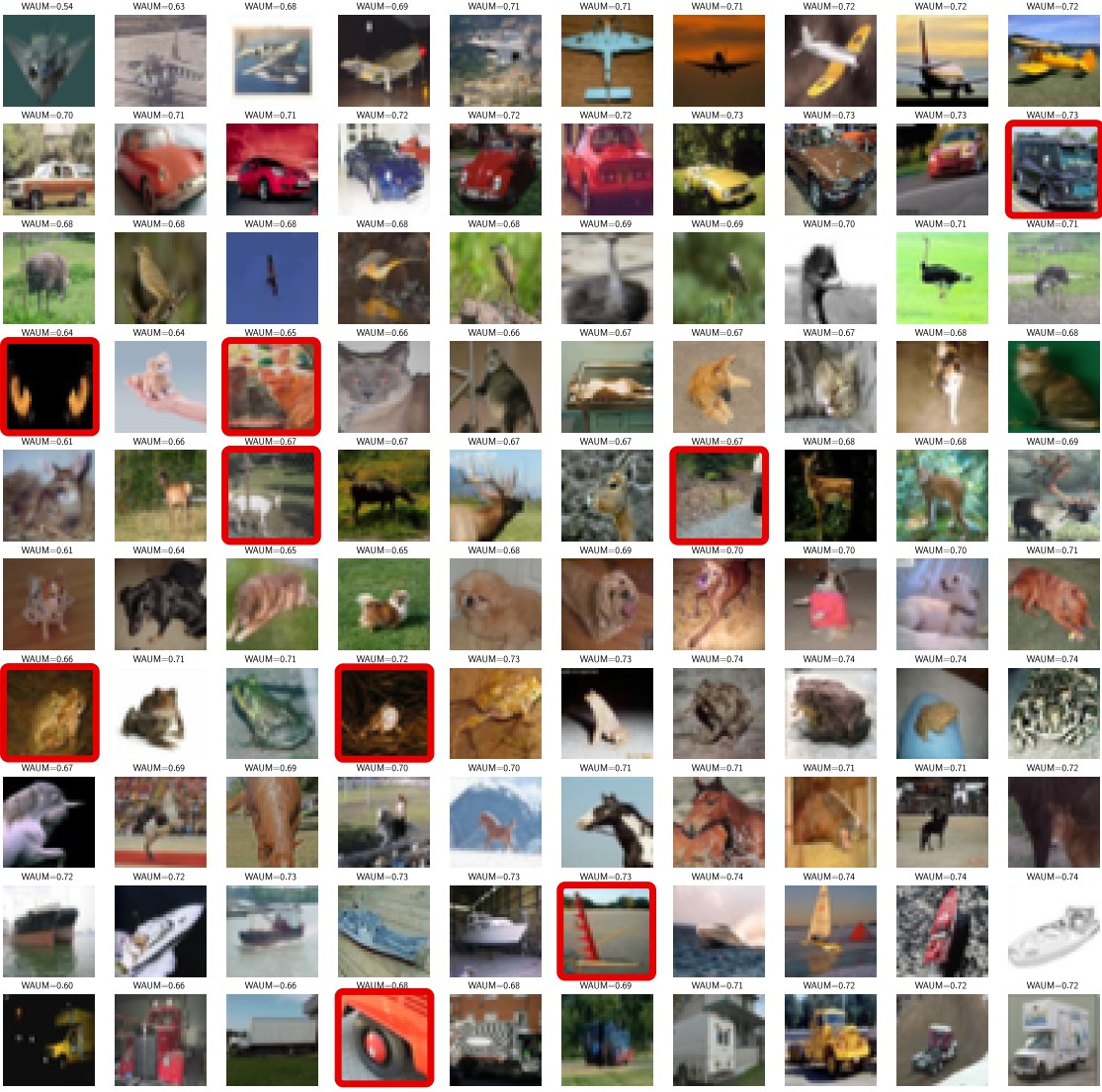

Figure 7: `CIFAR-10H`: 10 worst images for WAUM scores, by labels given in `CIFAR-10`. The rows represent the labels `airplane`, `automobile`, `bird`, `cat`, `deer`, `dog`, `frog`, `horse`, `ship`, and `truck`. Images in red can be particularly hard to classify as they are not typical examples of their label. Comparison with the AUMC and the AUM are available in Figure 16 Appendix D.2.1.

`CIFAR-10H` is a relatively well-curated dataset, and we observe in Section 4.2 that in this case, simple aggregation methods already perform well, in particular NS. Over the 2571 workers, less than 20 are identified as spammers using Raykar & Yu (2011) but note that most difficult tasks were removed when creating

the original `CIFAR-10` dataset. We refer to the *"labeler instruction sheet"* of Krizhevsky & Hinton (2009, Appendix C) for more information about the directives given to workers.

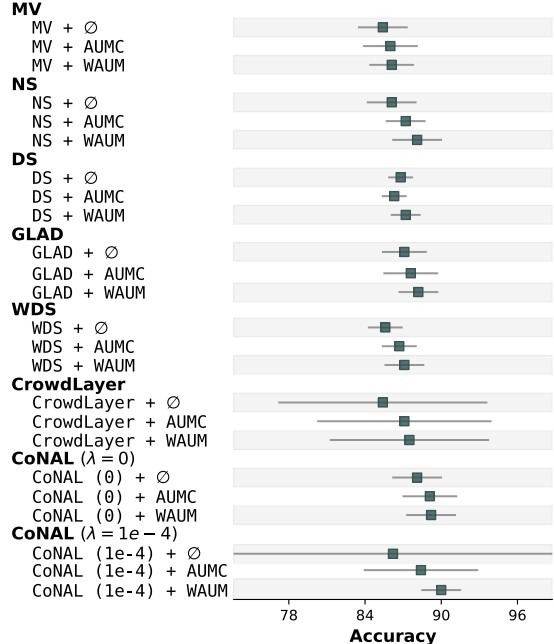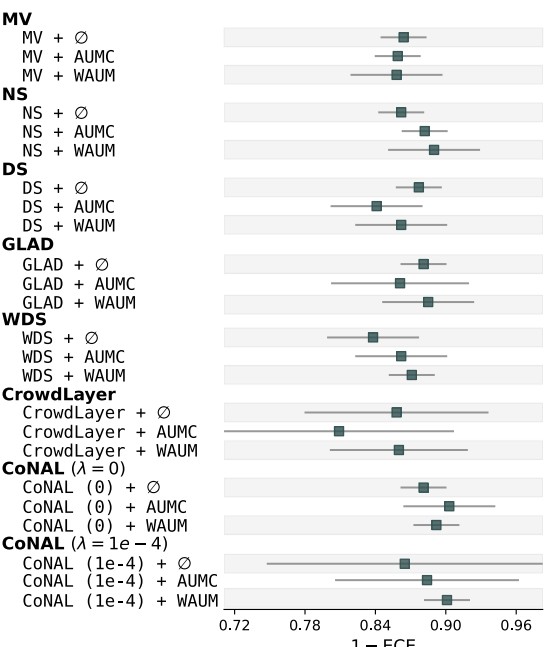

Figure 8: Ablation study on `LabelMe` using the VGG backbone: $\alpha = 0.01$. Errors are Gaussian confidence intervals at 95%. Numeric tables are available in Appendix F.

**LabelMe dataset.** This dataset consists in classifying 1000 images in $K = 8$ categories. In total 77 workers are reported in the dataset (though only 59 of them answered any task at all). Each task has between 1 and 3 labels. A validation set of 500 images and a test set of 1188 images are available.

We observe in Figure 8 that the WAUM improves the final test accuracy when combined with the CoNAL network with regularization. Note that the `LabelMe` dataset has classes that overlap and thus lead to intrinsic ambiguities. This is the reason why the CoNAL strategy was introduced by Chu et al. (2021): modeling common confusions help the network's decision, so it was expected for the CoNAL to perform well. Combined with our WAUM, additional gains are obtained on both metrics. The vanilla strategy, either for aggregation or learning, can be improved using a pruning preprocessing step. However, between the AUMC and the WAUM, we show a consistent improvement on using the WAUM that considers weights for the workers individually. For example, the classes `highway`, `insidecity`, `street` and `tallbuilding` (in rows) are overlapping for some tasks: some cities have streets with tall buildings, leading to confusion as shown in Figure 10.

**Music dataset.** This dataset differs from `LabelMe` and `CIFAR-10H` as it consists in classifying 1000 recordings of 30 seconds into $K = 10$ music genres. All the 44 workers involved voted for at least one music, resulting in up to 7 labels per task. Instead of classifying the original audio files, we use the associated Mel spectrograms following the methodology considered by Dong (2018) to retrieve an image classification setting. Though the benefits are not as striking as before on test accuracy, the ECE is slightly improved by combining our WAUM with CoNAL as can be seen in Table 6. Moreover, we show constant improvement of the test generalization performance using the WAUM preprocessing either in accuracy or in calibration.

Among other interesting discoveries, the `WAUM` helped us detect that the music *Zydeco Honky Tonk* by Buckwheat Zydeco was labeled as `classical`, `country` or `pop` by the workers, though it is a `blues` standard.

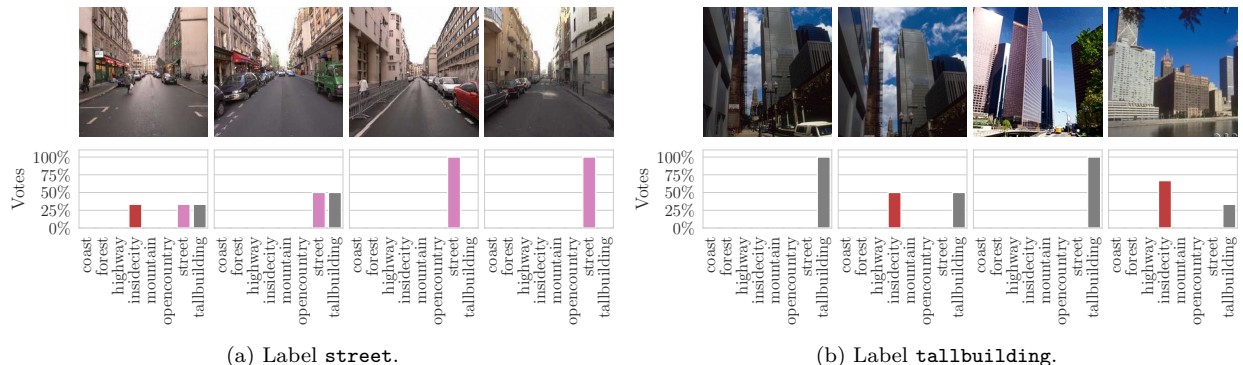

(a) Label `street`.      (b) Label `tallbuilding`.

Figure 9: `LabelMe` dataset: Worst WAUM for classes (top) and the associated voting distribution for each image (bottom). (a) Label `street` (b) Label `tallbuilding`. Even if the two tasks are very similar, because the workers are different the associated proposed labels can differ and add noise during training.

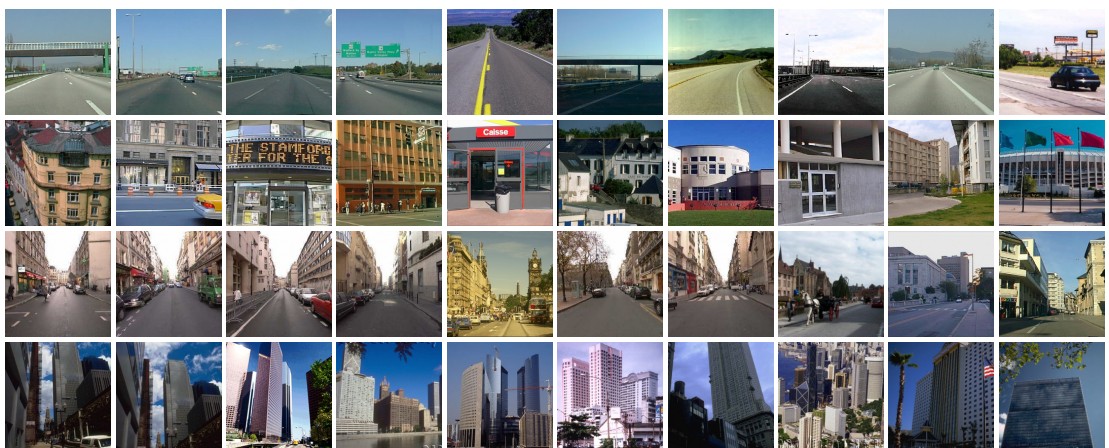

Figure 10: `LabelMe`: top-10 worst images detected by the WAUM (with labels row-ordered from top to bottom: `highway`, `insidecity`, `street`, `tallbuilding`). Overlapping classes lead to labeling confusion and learning difficulties for both the workers and the neural network.

Another example is *Caught in the middle* by Dio classified (with the same number of votes) as `rock`, `jazz`, or `country` though it is a `metal` song. One last example detected: the music *Patches* by Clarence Carter is stored in the `disco00020.wav` file. The true label is supposed to be `disco`, while the workers have provided the following labels: two have chosen `rock`, two `blues`, one `pop` and another one proposed `country`. The actual genre of this music is `country`-soul, so both the true label and five out of six workers are incorrect.

**WAUM sensitivity to the neural network architecture** In the following, we explore the architecture's impact on the generalization performance using the WAUM preprocessing. We compare three architectures, a VGG-16 with two dense layers added from Rodrigues & Pereira (2018), a Resnet-18 and a Resnet-34. We show in Figure 12 that depending on the network used, performance vary, but the WAUM step improves generalization performance in most cases (and does not worsen it).

**Limitations: computing the weights with many classes** First, concerning the weights $s_i^{(j)}$ (reflecting the trust in the image/worker interaction), we rely on confusion matrices $\{\hat{\pi}^{(j)}\}_{j \in [n_{\text{worker}}]}$. The DS model (Dawid & Skene, 1979) can be naturally used to estimate such matrices $\pi^{(j)} \in \mathbb{R}^{K \times K}$ for each worker $w_j$. Yet, the quadratic number of parameters (w.r.t. $K$) to be estimated for each worker can create convergence

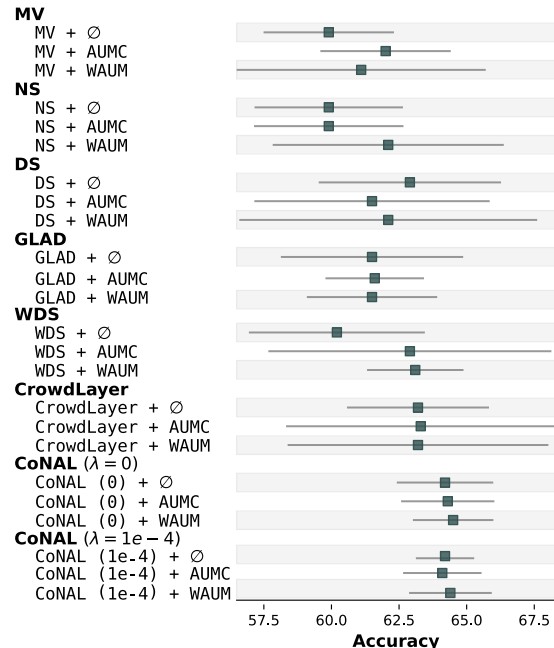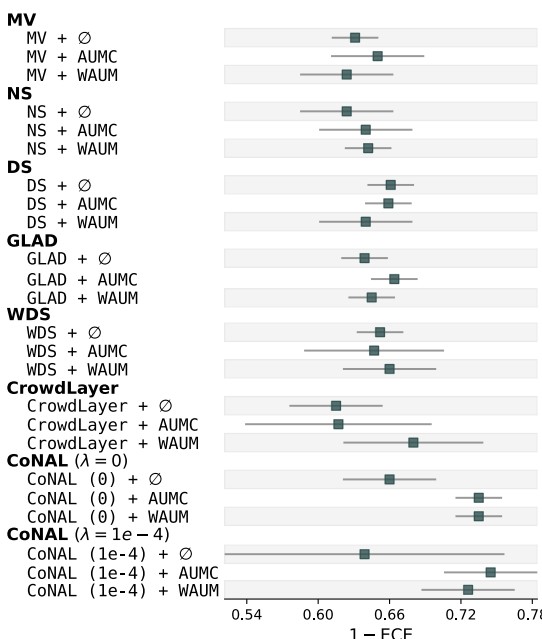

Figure 11: Ablation study on `Music` using the VGG backbone: $\alpha = 0.05$. Errors are Gaussian confidence intervals at 95%. Numeric tables are available in Appendix F.

issues for the vanilla DS model when $K$ is large. But as stated in Section 3, any model that can estimate confusion matrices can be considered for the WAUM's computation. We detail below some possible variants, that could help computing the confusion matrices used in the WAUM for the trust score computation.

- Sinha et al. (2018) accelerated the vanilla DS by constraining the estimated labels' distribution to be a Dirac mass. Hence, predicted labels are hard labels. This leads to worse calibration errors than vanilla DS but preserves the same accuracy.

- Passonneau & Carpenter (2014) introduced Dirichlet priors on the confusion matrices' rows and the prevalence $\rho$ to incorporate previously known information on the workers in the model (*e.g.,* from other experiments).

- Servajean et al. (2017) exploited the sparsity of the confusion matrices to cope with a large $K$.

- Imamura et al. (2018) estimated with variational inference $L \ll n_{\texttt{worker}}$ clusters of workers, constraining at most $L$ different confusion matrices. This reduces the number of parameters required from $K^2 \times n_{\texttt{worker}}$ to $K^2 \times L$.

**Pruning and *i.i.d* assumption**   For the pruning at preprocessing can induce a distortion in the training data distribution. A usual assumption made on learning problems is that the task/label pairs are *i.i.d.* However, by removing some of the hardest tasks, the new dataset $\mathcal{D}_{\text{pruned}}$ contains tasks that are not independent anymore. We should also keep in mind that Ilyas et al. (2022) have shown that in the standard datasets, the data is not *i.i.d* to begin with. Moreover, we should not set $\alpha$ too high as in imbalanced settings this might cause even more imbalance.

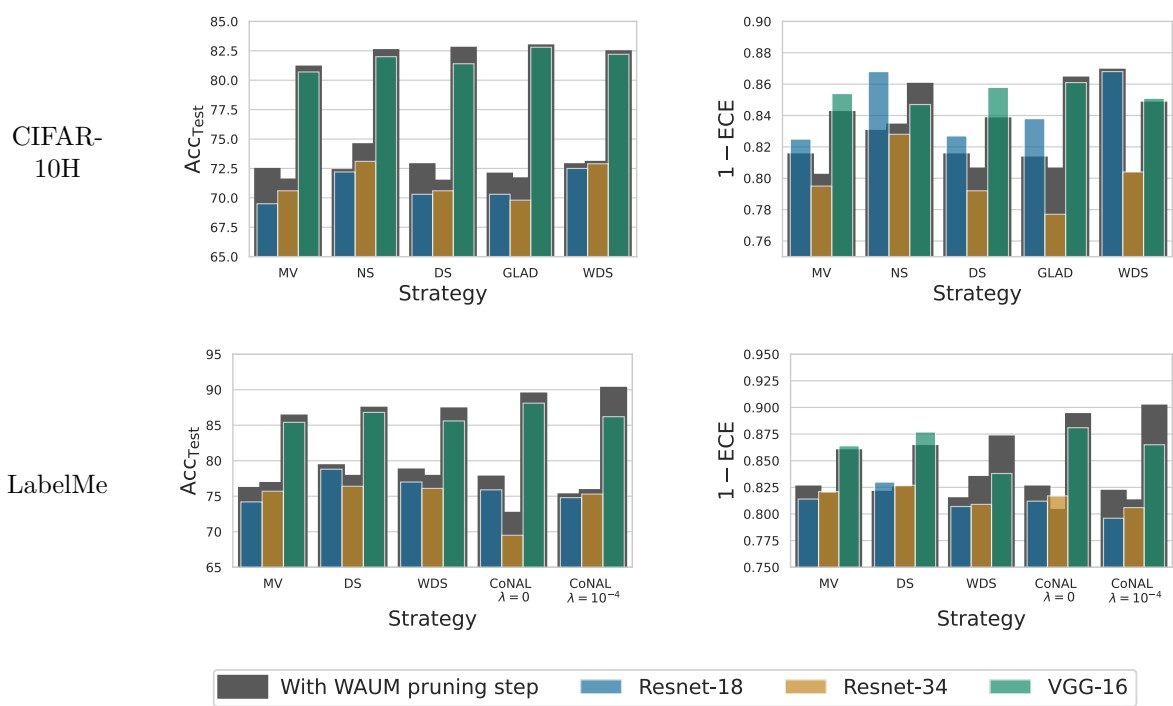

Figure 12: Performance obtained by training on the pruned dataset from the WAUM preprocessing step on `CIFAR-10H` and `LabelMe`. We consider multiple neural network architectures – ResNet-18, ResNet-34 or VGG-16 with batch normalization and two supplementary dense layers. We show that performance in accuracy are improved in most cases. Calibration performance in term of ECE fluctuate depending on the architecture considered, especially for the `CIFAR-10H` dataset. Using the WAUM with CoNAL on the `LabelMe` dataset, we obtain best performance both in accuracy and calibration.

## 5    Conclusion

In this paper, we investigate crowdsourcing aggregation models and how judging systems may impact generalization performance. Most models consider the ambiguity from the workers' perspective (very few consider the difficulty of the task itself) and evaluate workers on hard tasks that might be too ambiguous to be relevant, leading to a performance drop. Using a popular model (DS), we develop the WAUM, a flexible feature-aware metric that can identify hard tasks and improves generalization performance over vanilla strategies and naive pruning AUMC. It also yields a fair evaluation of workers' abilities and supports recent research on data pruning in supervised datasets. Independently of pruning, the WAUM allows identifying early the images that need extra labeling efforts or that are impossible to correctly label.

Extension of the WAUM to more general learning tasks (*e.g.,* top-$k$ classification, Appendix G) would be natural, including sequential label. Indeed, the WAUM could help to identify tasks requiring additional expertise and guide how to allocate more experts/workers for such identified tasks. Future works could adapt the WAUM to imbalanced crowdsourced datasets to identify potentially too ambiguous images that naturally occur in open platforms like Pl@ntNet[4]. And in this case, a class-dependent pruning threshold quantile could be used to avoid a learning bias for classes with very few instances.

Last but not least, on the dataset side, we believe that the community would benefit from releasing a challenging dataset (such as the one by Garcin et al. (2021) for instance) tailored to learn in crowdsourcing settings. Indeed, a dataset with the following properties could greatly foster future research in the field: a

---

[4]https://plantnet.org/en/

varying number of labels per worker, a high number of classes, and a subset with ground truth labels to test generalization performance.

### Broader Impact Statement

As this work proposes a method to prune tasks from training datasets based on human-derived data, we remind that pruning based on learning difficulty can induce a learning bias for the model. To mitigate this, only pruning a small portion of the dataset can help avoid any class with a small number of representatives to be removed of the dataset. Also, in this paper, we only remove tasks that are difficult to classify, we do not remove workers from the dataset. In particular, there is no repercussion on their pay, and by only evaluating them on tasks that are not detected as ambiguous, we evaluate their abilities on fairer tasks. Finally, during the entire procedure, all anonymity is conserved for workers, no other data than their anonymous identification number is used.

### Acknowledgments

This work was supported by the ANR through the Chaire IA CaMeLOt (ANR-20-CHIA-0001-01)

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

# A Popular label aggregation techniques

Several aggregation techniques can transform crowdsourced labels into probability distributions (soft labels). For any $d \in \mathbb{N}$ and $z \in (0, \infty)^d$, let $\mathrm{Norm}(z) \in (0, \infty)^d$ be the vector defined by $\forall i \in [d]$, $\mathrm{Norm}(z)_i = z_i / \sum_{i'=1}^d z_{i'}$.

## A.1 Naive soft (NS)

The naive soft (NS) labeling is simply the empirical distribution of the answered votes:

$$\forall x_i \in \mathcal{D}_{\text{train}}, \quad \hat{y}_i^{\text{NS}} = \mathrm{Norm}(\tilde{y}_i), \quad \text{where } \tilde{y}_i = \Big( \sum_{j \in \mathcal{A}(x_i)} \mathbb{1}_{\{y_i^{(j)} = k\}} \Big)_{k \in [K]} . \tag{6}$$

## A.2 Majority voting (MV)

Majority voting (MV) outputs the most answered label:

$$\forall x_i \in \mathcal{D}_{\text{train}}, \quad \hat{y}_i^{\text{MV}} = \underset{k \in [K]}{\arg\max} \Big( \sum_{j \in \mathcal{A}(x_i)} \mathbb{1}_{\{y_i^{(j)} = k\}} \Big) . \tag{7}$$

## A.3 Dawid and Skene (DS)

The Dawid and Skene (Dawid & Skene, 1979) model aggregates answers and evaluates the workers' confusion matrix to observe where their expertise lies exactly. Let us introduce $\rho_\ell$ the prevalence of each label in the dataset (*i.e.,* $\mathbb{P}(y_i^\star = \ell)$), the probability that a task drawn at random is labeled $\ell \in [K]$. Following standard notations, we also write $\{T_{i,\ell}, \ i \in [n_{\text{task}}]\}$ the indicator variables for task $i$, that is $T_{i,\ell} = 1$ if the true label for task $i$ is $\ell$ (*i.e.,* $y_i^\star = \ell$) and zero otherwise. Finally, let $\pi_{\ell,k}^{(j)}$ be the probability for worker $j$ to select label $k$ when $y^\star = \ell$. The model's likelihood reads:

$$\prod_{i \in [n_{\text{task}}]} \prod_{\ell \in [K]} \left[ \rho_\ell \prod_{j \in [n_{\text{worker}}]} \prod_{k \in [K]} \left( \pi_{\ell,k}^{(j)} \right)^{\mathbb{1}_{\{y_i^{(j)} = k\}}} \right]^{T_{i\ell}} . \tag{8}$$

To maximize the likelihood, we use the EM algorithm (Dempster et al., 1977) to estimate the parameters $\pi_{\ell,k}^{(j)}$ and $\rho_\ell$, using $(T_{i,\cdot})_{i \in [n_{\text{task}}]}$ as latent variables. Our implementation of the EM algorithm is given in Algorithm 2. The convergence criterion we use in practice is that the likelihood has not decreased more than $\epsilon > 0$ between two iterations. By default, $\epsilon$ is set to $10^{-6}$, and the EM algorithm stops at iteration $t \in \mathbb{N}$ if $\left| \mathrm{Likelihood}_t - \mathrm{Likelihood}_{t+1} \right| < \varepsilon$.

## A.4 Weighted Dawid and Skene (WDS)

Let us run the DS model to get estimated confusion matrices $\hat{\pi}^{(j)} \in \mathbb{R}^{K \times K}$ for $j \in [n_{\text{worker}}]$. Now, remind that for a given worker $j \in [n_{\text{worker}}]$ and a class $k \in [K]$, the term $\hat{\pi}_{k,k}^{(j)}$ estimate the probability for worker $w_j$ to recognize a task whose true label is $k$. We use this term as a trust score and define the WDS soft label as

$$\forall x_i \in \mathcal{D}_{\text{train}}, \quad \hat{y}_i^{\text{WDS}} = \mathrm{Norm}(\tilde{y}_i), \quad \text{with} \quad \tilde{y}_i = \Big( \sum_{j \in \mathcal{A}(x_i)} \hat{\pi}_{k,k}^{(j)} \mathbb{1}_{\{y_i^{(j)} = k\}} \Big)_{k \in [K]} . \tag{9}$$

## A.5 Generative model of Labels, Abilities, and Difficulties (GLAD)

We recall the GLAD (Whitehill et al., 2009) algorithm in the binary setting. A modeling assumption is that the $j$-th worker labels correctly the $i$-th task with probability given by

$$\mathbb{P}(y_i^{(j)} = y_i^\star | \alpha_j, \beta_i) = \frac{1}{1 + e^{-\alpha_j \beta_i}} , \tag{10}$$

---

**Algorithm 2** DS (EM version)

---

**Input**: $\mathcal{D}_{\text{train}}$: crowdsourced dataset

**Output**: $(\hat{y}_i^{\text{DS}})_{i \in [n_{\text{task}}]} = (\hat{T}_{i,\cdot})_{i \in [n_{\text{task}}]}$: estimated soft labels and $\{\hat{\pi}^{(j)}\}_{j \in [n_{\text{worker}}]}$: estimated confusion matrices

1: **Initialization:** $\forall i \in [n_{\text{task}}], \forall \ell \in [K], \ \hat{T}_{i,\ell} = \frac{1}{|\mathcal{A}(x_i)|} \sum_{j \in \mathcal{A}(x_i)} \mathbb{1}_{\{y_i^{(j)}=\ell\}}$

2: **while** Likelihood not converged **do**

3:     Get $\hat{\pi}$ and $\hat{\rho}$ assuming $\hat{T}$s are known

4:     $\forall (\ell, k) \in [K]^2, \ \hat{\pi}_{\ell,k}^{(j)} \leftarrow \dfrac{\sum_{i \in [n_{\text{task}}]} \hat{T}_{i,\ell} \cdot \mathbb{1}_{\{y_i^{(j)}=k\}}}{\sum_{k' \in [K]} \sum_{i' \in [n_{\text{task}}]} \hat{T}_{i',\ell} \cdot \mathbb{1}_{\{y_{i'}^{(j)}=k'\}}}$

5:     $\forall \ell \in [K], \ \hat{\rho}_\ell \leftarrow \frac{1}{n_{\text{task}}} \sum_{i \in [n_{\text{task}}]} \hat{T}_{i,\ell}$

6:     Estimate $\hat{T}$s knowing $\hat{\pi}$ and $\hat{\rho}$

7:     $\forall (i, \ell), \in [n_{\text{task}}] \times [K], \hat{T}_{i\ell} \leftarrow \dfrac{\prod_{j \in \mathcal{A}(x_i)} \prod_{k \in [K]} \hat{\rho}_\ell \cdot \left(\hat{\pi}_{\ell,k}^{(j)}\right)^{\mathbb{1}_{\{y_i^{(j)}=k\}}}}{\sum_{\ell' \in [K]} \prod_{j' \in \mathcal{A}(x_i)} \prod_{k' \in [K]} \hat{\rho}_{\ell'} \cdot \left(\hat{\pi}_{\ell'k'}^{(j')}\right)^{\mathbb{1}_{\{y_i^{(j')}=k'\}}}}$

8: **end while**

---

**Algorithm 3** GLAD (EM version)

---

**Input**: $\mathcal{D}_{\text{train}}$: crowdsourced dataset

**Output**: $\alpha = \{\alpha_j\}_{j \in [n_{\text{worker}}]}$: worker abilities, $\beta = \{\beta_i\}_{i \in [n_{\text{task}}]}$: task difficulties, aggregated labels

1: **while** Likelihood not converged **do**

2:     Estimate probability of $y_i^\star$

3:     $\forall i \in [n_{\text{task}}], \ \mathbb{P}(y_i^\star | \{y_i^{(j)}\}_i, \alpha, \beta_i) \propto \mathbb{P}(y_i^\star) \prod_j \mathbb{P}(y_i^{(j)} | y_i^\star, \alpha_j, \beta_i)$

4:     Maximization step

5:     Maximize auxiliary function $Q(\alpha, \beta)$ in Equation (11) w.r.t. $\alpha$ and $\beta$

6: **end while**

---

with $\alpha_j \in \mathbb{R}$ the worker's expertise: $\alpha_j < 0$ implies misunderstanding, $\alpha_j = 0$ an impossibility to separate the two classes and $\alpha_j > 0$ a valuable expertise. The coefficient $1/\beta_i \in \mathbb{R}_+$ represents the task's intrinsic difficulty: if $1/\beta_i \to 0$ the task is trivial; on the other side when $1/\beta_i \to +\infty$ the task is very ambiguous. Parameters $(\alpha_j)_{j \in [n_{\text{worker}}]}$ and $(\beta_i)_{i \in [n_{\text{task}}]}$ are estimated using an EM algorithm as described in Algorithm 3.

The auxiliary function for the binary GLAD model is:

$$Q(\alpha, \beta) = \mathbb{E}[\log \mathbb{P}(\{y_i^{(j)}\}_{ij}, \{y_i^\star\}_i)] = \sum_i \mathbb{E}[\log \mathbb{P}(y_i^\star)] + \sum_{ij} \mathbb{E}[\log \mathbb{P}(y_i^{(j)} | y_i^\star, \alpha_j, \beta_i)] \ . \tag{11}$$

An extension to the multiclass setting is given by Whitehill et al. (2009) under the following assumption: the distribution over all incorrect labels is supposed uniform. In this setting, the model assumption from Equation (10) still holds and

$$\forall k \neq y_i^\star, \ \mathbb{P}(y_i^{(j)} = k | \alpha_j, \beta_i) = \frac{1}{K-1} \left(1 - \frac{1}{1 + e^{-\alpha_j \beta_i}}\right) \ .$$

However, this is not verified in many practical cases, as can be seen for example in Figure 2c where the `cat` label is only mistaken `deer` and not with other ones. We have used the implementation from `https://github.com/notani/python-glad` to evaluate the GLAD performance in our experiments. The maximization of the function $Q$ w.r.t. $\alpha$ and $\beta$ is performed using a conjugate gradient solver. The initial parameters are all set to 1.

## A.6 CrowdLayer and its matrix weights strategy (MW)

From (Rodrigues & Pereira, 2018), CrowdLayer is an end-to-end strategy in the crowdsourcing setting. From the output of a neural network, a new layer called *crowd layer* is added to take into account worker

specificities. The main classifier thus becomes globally shared, and the new layer is the only worker-aware layer. As multiple variants of CrowdLayer can exist, we only considered in this paper the matrix weights (MW) strategy that is akin to the DS model. Denoting $z = f(x_i)$ the output of the neural network classifier $f$ for a given task $x_i$ labeled by a worker $w_j$, the added layer multiplies $z$ by a matrix of weights $W^j \in \mathbb{R}^{K \times K}$. This matrix of weights per worker takes into account the local confusion of each worker. In practice, the forward pass $F$ on a task $x_i$ annotated by worker $w_j$ using CrowdLayer computes $F(x_i, w_j) = W^j \sigma(f(x_i))$.

## A.7 Common Noise Adaptation Layers (CoNAL)

CrowdLayer takes into account worker-specific confusion matrices. CoNAL (Chu et al., 2021) generalizes this setting by creating a global confusion matrix $W^g \in \mathbb{R}^{K \times K}$ in addition to the local ones $W^j \in \mathbb{R}^{K \times K}$ for $j \in [n_{\texttt{worker}}]$ working all together with the classifier $f$. Given a worker $w_j$, the confusion is global with weight $\omega_i^j$ and local with weight $1 - \omega_i^j$. The final distribution output used to compute the loss is given by:

$$p_{\text{out}}(x_i, w_j) = \omega_i^j W^g f(x_i) + (1 - \omega_i^j) W^j f(x_i) \ .$$

As is, CoNAL local matrices tend to aggregate themselves onto the global matrix. To avoid this phenomenon, a regularization term in the loss can be added as leading to the final loss:

$$\mathcal{L}(W^g, \{W^j\}_{j \in [n_{\texttt{worker}}]}) = \frac{1}{n_{\texttt{task}}} \sum_{i \in [n_{\texttt{task}}]} \sum_{j \in [n_{\texttt{worker}}]} \mathrm{H}\Big(y_i^{(j)}, p_{\text{out}}(x_i, w_j)\Big) - \lambda \sum_{j \in [n_{\texttt{worker}}]} \|W^g - W^j\|_2 \ ,$$

with $\lambda$ the regularization hyperparameter and H the crossentropy loss. The larger $\lambda$, the farther local confusion weights are from the shared confusion.

# B AUM and WAUM additional details

---
**Algorithm 4** worker-wise WAUM.

---
1: **Input:** $\mathcal{D}_{\text{train}}$: tasks and crowdsourced labels, $\alpha \in [0, 1]$: proportion of training points pruned, $T \in \mathbb{N}$: number of epochs, Est: Estimation procedure for the confusion matrices
2: **Initialization:** Get confusion matrices $\{\hat{\pi}^{(j)}\}_{j \in [n_{\texttt{worker}}]}$ from Est (= DS by default)
3: **for** $j \in [n_{\texttt{worker}}]$ **do**
4:     **for** $T$ epochs **do**
5:         **Train** a neural network for $T$ epochs on $\mathcal{D}_{\text{train}}^{(j)} = \Big\{ \big(x_i, y_i^{(j)}\big) \text{ for } i \in \mathcal{T}(w_j)\Big\}$
6:     **end for**
7:     Get $\text{AUM}(x_i, y_i^{(j)}; \mathcal{D}_{\text{train}}^{(j)})$ using Equation (2)
8:     Get **trust scores** $s^{(j)}(x_i)$ using Equation (5)
9: **end for**
10: **for** each task $x \in \mathcal{X}_{\text{train}}$ **do**
11:     Compute $\text{WAUM}(x)$ using Equation (4)
12: **end for**
13: Get $q_\alpha$ the **quantile threshold** of order $\alpha$ of $(\text{WAUM}(x_i))_{i \in [n_{\texttt{task}}]}$
14: Define $\mathcal{D}_{\text{pruned}} = \Big\{ \big(x_i, (y_i^{(j)})_{j \in \mathcal{A}(x_i)}\big) : \text{WAUM}(x_i) \geq q_\alpha \text{ for } i \in [n_{\texttt{task}}]\Big\}$

---

## B.1 Unstacking workers answers in the WAUM: the worker-wise WAUM

In Algorithm 1, the WAUM requires training a classifier directly from all votes. If the crowdsourcing experiment generates many answers per worker, for example when each worker answers all the tasks, we can modify Algorithm 1 to train one classifier per worker for $T$ epochs instead of a single one *à la* Guan et al. (2017). This means that each classifier is only trained on $\mathcal{D}^{(j)} := \{(x_i, y_i^{(j)})\}_{i \in [n_{\texttt{task}}]}$ to compute the AUM of the tasks answered. We refer to this as the *worker-wise* WAUM and give the full algorithm in Algorithm 4.

By doing so, the network trained for a given worker is not influenced by the answers of the other workers. Hence, the AUM computed by this worker-wise WAUM is independent across workers (assuming workers are answering independently). One downside of this worker-wise application is its training cost that increases drastically. Where the vanilla WAUM adds a cost of $T$ epochs before training to identify ambiguous tasks, *worker-wise* WAUM adds a cost of $T \times n_{\texttt{worker}}$ epochs.

In the simulated examples we propose, we provide the results for the worker-wise WAUM, yet in such simulated cases with many labels per task, the results do not differ much from the WAUM; see for instance Table 4.

### B.2 AUM computation in practice.

We recall in Algorithm 5 how to compute the AUM in practice for a given training set $\mathcal{D}_{\text{train}}$. This step is used within the WAUM (label aggregation step). Overall, w.r.t. training a model, computing the AUM requires an additional cost: $T$ training epochs are needed to record the margins' evolution for each task. This usually represents less than twice the original time budget. We recall that $\sigma^{(t)}(x_i)$ is the softmax output of the predicted scores for the task $x_i$ at iteration $t$.

---

**Algorithm 5** AUM algorithm

---

**Input:** $\mathcal{D}_{\text{train}} = (x_i, y_i)_{i \in [n_{\texttt{task}}]}$: training set with $n_{\texttt{task}}$ task/label couples, $T \in \mathbb{N}$: number of epochs
**for** $t = 1, \ldots, T$ **do**
    **Train** the neural network for the $t^{th}$ epoch, using $\mathcal{D}_{\text{train}}$
    **for** $i \in [n_{\texttt{task}}]$ **do**
        **Record softmax** output $\sigma^{(t)}(x_i) \in \Delta_{K-1}$
        **Compute margin** $M^{(t)}(x_i, y_i) = \sigma_{y_i}^{(t)}(x_i) - \sigma_{[2]}^{(t)}(x_i)$
    **end for**
**end for**
$\forall i \in [n_{\texttt{task}}], \ \text{AUM}(x_i, y_i; \mathcal{D}_{\text{train}}) = \frac{1}{T} \sum_{t \in [T]} M^{(t)}(x_i, y_i)$

---

## C   Reminder on the calibration of neural networks

Hereafter, we propose a reminder on neural networks calibration metric defined in Guo et al. (2017). Calibration measures the discrepancy between the accuracy and the confidence of a network. In this context, we say that a neural network is perfectly calibrated if it is as accurate as it is confident. For each task $x \in \mathcal{X}_{\text{train}} = \{x_1, \ldots, x_{n_{\texttt{task}}}\}$, let us recall that an associated predicted probability distribution is provided by $\sigma(x) \in \Delta_{K-1}$. Let us split the prediction interval $[0, 1]$ into $M = 15$ bins $I_1, \ldots, I_M$ of size $1/M$: $I_m = (\frac{m-1}{M}, \frac{m}{M}]$, where $m = 1, \ldots, M$. Following Guo et al. (2017), we denote $B_m = \{x \in \mathcal{X}_{\text{train}} : \sigma_{[1]}(x) \in I_m\}$ the task whose predicted probability is in the $m$-th bin[5]. We recall that the accuracy of the network for the samples in $B_m$ is given by $\text{acc}(B_m)$ the empirical confidence by $\text{conf}(B_m)$:

$$\text{acc}(B_m) = \frac{1}{|B_m|} \sum_{i \in B_m} \mathbb{1}_{\{\sigma_{[1]}(x_i) = y_i\}} \quad \text{and} \quad \text{conf}(B_m) = \frac{1}{|B_m|} \sum_{i \in B_m} \sigma_{[1]}(x_i) \ .$$

Finally, the expected calibration error (ECE) reads:

$$\text{ECE} = \sum_{m=1}^{M} \frac{|B_m|}{n_{\texttt{task}}} |\text{acc}(B_m) - \text{conf}(B_m)| \ . \tag{12}$$

A neural network is said *perfectly calibrated* if $\text{ECE} = 0$, thus if the accuracy equals the confidence for each subset $B_m$.

---

[5]Remember that with our notation $\sigma_{[1]}(x) = \arg\max_{k \in [K]} (\sigma(x))_k$, with ties broken at random.

# D  Datasets description

## D.1  Synthetic dataset

In this section, we present simulated datasets to showcase the specificities and possible limitations of the WAUM. Here is a summary of the experiments detailed in the following sub-sections:

1. **The `three_circles` dataset**: we explain further how the simulations in Section 4 were conducted

2. **The `two_moons` dataset**: we showcase a setting where the ambiguous tasks should be kept and not pruned. No simulated worker was able to get past the intrinsic difficulty of the dataset.

3. **The `make_classication_many_workers` dataset**: we showcase a setting with many workers and few labels per task. In this case, it is more relevant to consider the WAUM instead of the worker-wise WAUM.

### D.1.1  The `three_circles` dataset

This dataset was presented in Section 4, we give additional details here. We simulate three cloud points using `scikit-learn`'s function `two_circles`. Each of the $n_{\texttt{task}} = 525$ points represents a task. The $n_{\texttt{worker}} = 3$ workers are standard classifiers: $w_1$ is a linear Support Vector Machine Classifier (linear SVC), $w_2$ is an SVM with RBF kernel (SVC), and $w_3$ is a gradient boosted classifier (GBM) with five estimators. To induce more ambiguity (and avoid too similar workers), the SVC has a maximum iteration set to 1 in the learning phase. Other hyperparameters are set to `scikit-learn`'s default values[6]. Data is split between train (70%) and test (30%) and each simulated worker votes for each task, *i.e.,* for all $x \in \mathcal{X}_{\text{train}}$, $|\mathcal{A}(x)| = n_{\texttt{worker}} = 3$. The disagreement area is identified in the northeast area of the dataset as can be seen in Figure 4. Section 4.1 also shows that pruning too little data ($\alpha$ small) or too much ($\alpha$ large) can mitigate the performance.

### D.1.2  The `two_moons` dataset

This dataset is introduced as a case where pruning is not recommended, to illustrate the limitations of the worker-wise WAUM method. The `two_moons` simulation framework showcases the difference between relevant ambiguity in a dataset and an artificial one. This dataset is created using `make_moons` function from `scikit-learn`. We simulate $n_{\texttt{task}} = 500$ points, a noise $\varepsilon = 0.2$ and use a test split of 0.3.

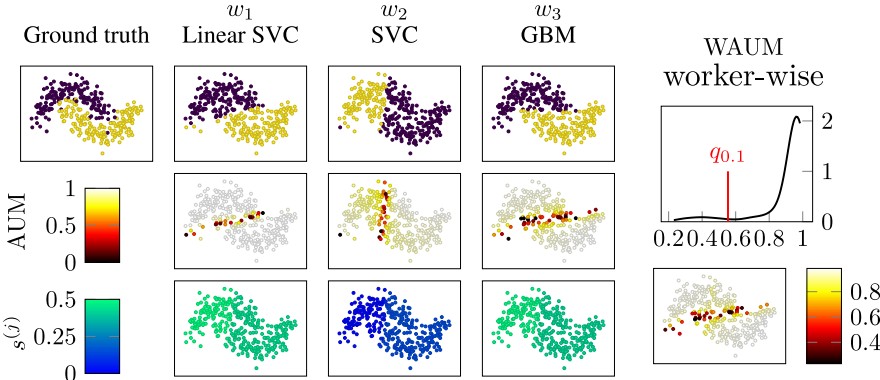

Figure 13: `two_moons` dataset: simulated workers with associated AUM and normalized trust scores. The hyperparameter $\alpha$ is set to 0.1 for the worker-wise WAUM. Notice that the SVC classifier is mostly wrong (since we only train for one epoch for this worker), inducing a lower trust score overall.

As can be observed with Figure 13 and Figure 14, the difficulty of this dataset comes from the two shapes leaning into one another. However, this intrinsic difficulty is not due to noise but is inherent to the data.

---

[6]For instance, the squared-hinge is penalized with an $\ell^2$ regularization parameter set to 1 for linear SVC and SVC, GBM uses as loss the multinomial deviance, and the maximum depth equals to 3 (default).

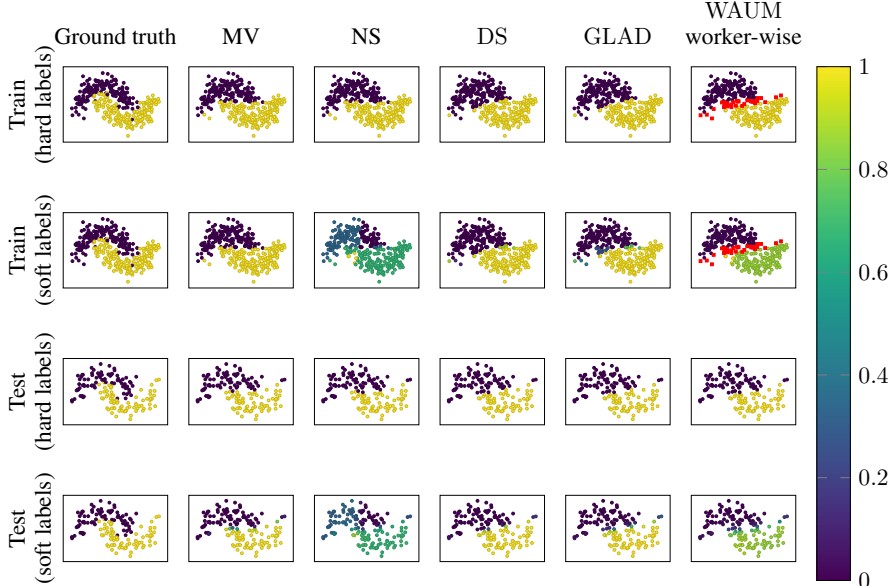

Figure 14: `two_moons` dataset: One realization of Table 3 varying the aggregation strategy. Label predictions on train/test sets provided by a three dense layers' artificial neural network $(30, 20, 20)$ trained on smooth labeled obtained by after aggregating the crowdsourced labels (as in Figure 13). Points in red are pruned from the training set in the worker-wise WAUM aggregation. The $\alpha$ hyperparameter is set to 0.1. Each point represents a task $x_i$, and its color is the probability to belong in class 1. One can visualize the ambiguity in the soft training aggregated labels, but also in the resulting predictions by the neural network. Errors represented are standard deviations.

Table 3: Training and test accuracy depending on the aggregation method used for the `two_moons`'s dataset with $n_{\texttt{task}} = 500$ points used for training a three dense layers' artificial neural network $(30, 20, 20)$. For reference, the best worker is $w_3$ with a training accuracy of 0.923 and a test accuracy of 0.900.

| Aggregation | $\mathrm{Acc}_{\mathrm{test}}$ | ECE |
|---|---|---|
| MV | $\mathbf{0.894 \pm 0.002}$ | $\mathbf{0.098} \pm 0.004$ |
| NS | $0.887 \pm 0.002$ | $0.217 \pm 0.010$ |
| DS | $0.867 \pm 0.000$ | $0.126 \pm 0.001$ |
| GLAD | $0.872 \pm 0.006$ | $0.107 \pm 0.004$ |
| worker-wise WAUM($\alpha = 10^{-3}$) | $0.875 \pm 0.002$ | $0.088 \pm 0.012$ |
| worker-wise WAUM($\alpha = 10^{-2}$) | $0.874 \pm 0.002$ | $0.092 \pm 0.011$ |
| worker-wise WAUM($\alpha = 10^{-1}$) | $0.870 \pm 0.003$ | $0.101 \pm 0.020$ |
| worker-wise WAUM($\alpha = 0.25$) | $0.829 \pm 0.006$ | $0.135 \pm 0.011$ |

In this case, removing the hardest tasks means removing points at the edges of the crescents, and those are important in the data's structure. From Table 3, we observe that learning on naive soft labeling leads to better performance than other aggregations. But with these workers, no aggregation produced labels capturing the shape of the data.

### D.1.3 The `make_classification_many_workers` dataset

We simulate $n_w = 150$ workers who answer tasks from a dataset with $K = 4$ classes simulated using `scikit-learn`'s function `make_classification`. In this setting, the WAUM has the same performance as the worker-wise WAUM, with a **much lower computational cost** (as we do not train $n_{\mathrm{worker}}$ networks but a single one). All simulated tasks are labeled by up to five workers among Linear SVCs, SVCs or Gradient

Boosted Classifiers (GBM) chosen uniformly. To simulate multiple workers with some dissimilarities, we randomly assign hyperparameters for each classifier as follows.

Each Linear SVC has a margin `C` chosen in a linear grid of 20 points from $10^{-3}$ to 3, a maximum number of iterations between 1 and 100, and either `hinge` or `squared_hinge` as `loss` function. Each SVC has a `poly` (with degree 3), `rbf` or `sigmoid` kernel and a maximum number of iterations between 1 and 100. Finally, each GBM has a learning rate of 0.01, 0.1 or 0.5, a given number of base estimators in $\{1, 2, 5, 10, 15, 20, 30, 50, 100\}$ and a maximum number of iterations between 1 and 100. All simulated workers are also initialized using different seeds. All hyperparameters are drawn uniformly at random from their respective set of possible values.

Table 4: The `make_classification_many_workers` dataset: Performance metrics by aggregation method. The number of tasks is $n_{\texttt{task}} = 250$ tasks per classes and $1 \le |\mathcal{A}(x)| \le 5$. Errors represented are standard deviations.

| Aggregation | $\text{Acc}_{\text{test}}$ | ECE |
|---|---|---|
| NS | $0.851 \pm 0.00$ | $0.146 \pm 0.023$ |
| DS | $0.849 \pm 0.004$ | $0.242 \pm 0.011$ |
| GLAD | $0.842 \pm 0.002$ | $0.196 \pm 0.004$ |
| worker-wise WAUM($\alpha = 10^{-1}$) | $0.849 \pm 0.006$ | $\mathbf{0.137} \pm 0.034$ |
| WAUM($\alpha = 10^{-1}$) | $\mathbf{0.861} \pm 0.007$ | $0.156 \pm 0.023$ |

## D.2 Real datasets

The datasets we consider are all decomposed into three parts: train ($\mathcal{D}_{\text{train}}$), validation ($\mathcal{D}_{\text{val}}$), and test ($\mathcal{D}_{\text{test}}$). They are described in the following subsections. In particular, we provide for the training set of each dataset (see Figures 15, 17 and 18) three visualizations: the feedback effort per task distribution ($|\mathcal{A}(x)|$), the load per worker distribution ($|\mathcal{W}(x)|$), and the naive soft labels entropy distribution, *i.e.,* the entropy distribution for each task in the training set, defined by: $\forall x_i \in \mathcal{X}_{\text{train}}$, $\text{Ent}(x_i) = -\sum_{k \in [K]} (\hat{y}_i^{\text{NS}})_k \log((\hat{y}_i^{\text{NS}})_k)$.

We have conducted experiments on three real datasets. The `CIFAR-10H` dataset has been proposed to reflect human perceptual uncertainty in (a subpart of) the classical `CIFAR-10` dataset. Each worker has annotated a large number of (seemingly easy) tasks, thus leading to few disagreements. The `LabelMe` and `Music` datasets have very few votes per task, leading to more ambiguous votes distributions.

### D.2.1 The `CIFAR-10H` dataset

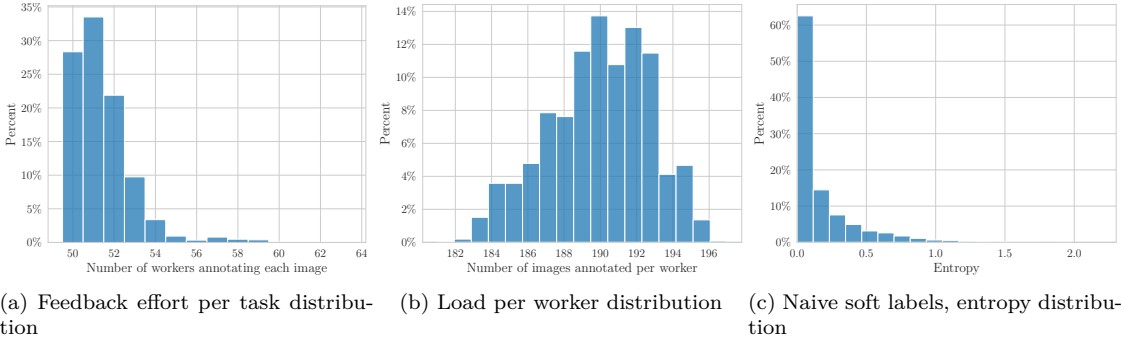

(a) Feedback effort per task distribution

(b) Load per worker distribution

(c) Naive soft labels, entropy distribution

Figure 15: `CIFAR-10H`: dataset visualization

Introduced by Peterson et al. (2019), the crowdsourced dataset `CIFAR-10H` attempts to recapture the human labeling noise present when creating the dataset. We have transformed this dataset, mainly by creating a

validation set. Hence, the training set for our version of `CIFAR-10H` consists of the first 9500 test images from `CIFAR-10`, hence $|\mathcal{D}_{\text{train}}| = 9500$. The validation set is then composed of the last 500 images from the training set of `CIFAR-10` meaning $|\mathcal{D}_{\text{test}}| = 500$. The test set consists of the whole training set from `CIFAR-10`, so $|\mathcal{D}_{\text{test}}| = 50000$. The crowdsourcing experimentation involved $n_{\text{worker}} = 2571$ workers on Amazon Mechanical Turk. Workers had to choose one label for each presented image among the $K = 10$ labels of `CIFAR-10`: `airplane`, `automobile`, `bird`, `cat`, `deer`, `dog`, `frog`, `horse`, `ship` and `truck`. Each worker labeled 200 tasks (and was paid \$1.50 for that): 20 for each original category. Answering time was also measured for each worker[7]. The `CIFAR-10H` annotating effort is balanced: each task has been labeled by 50 workers on average.

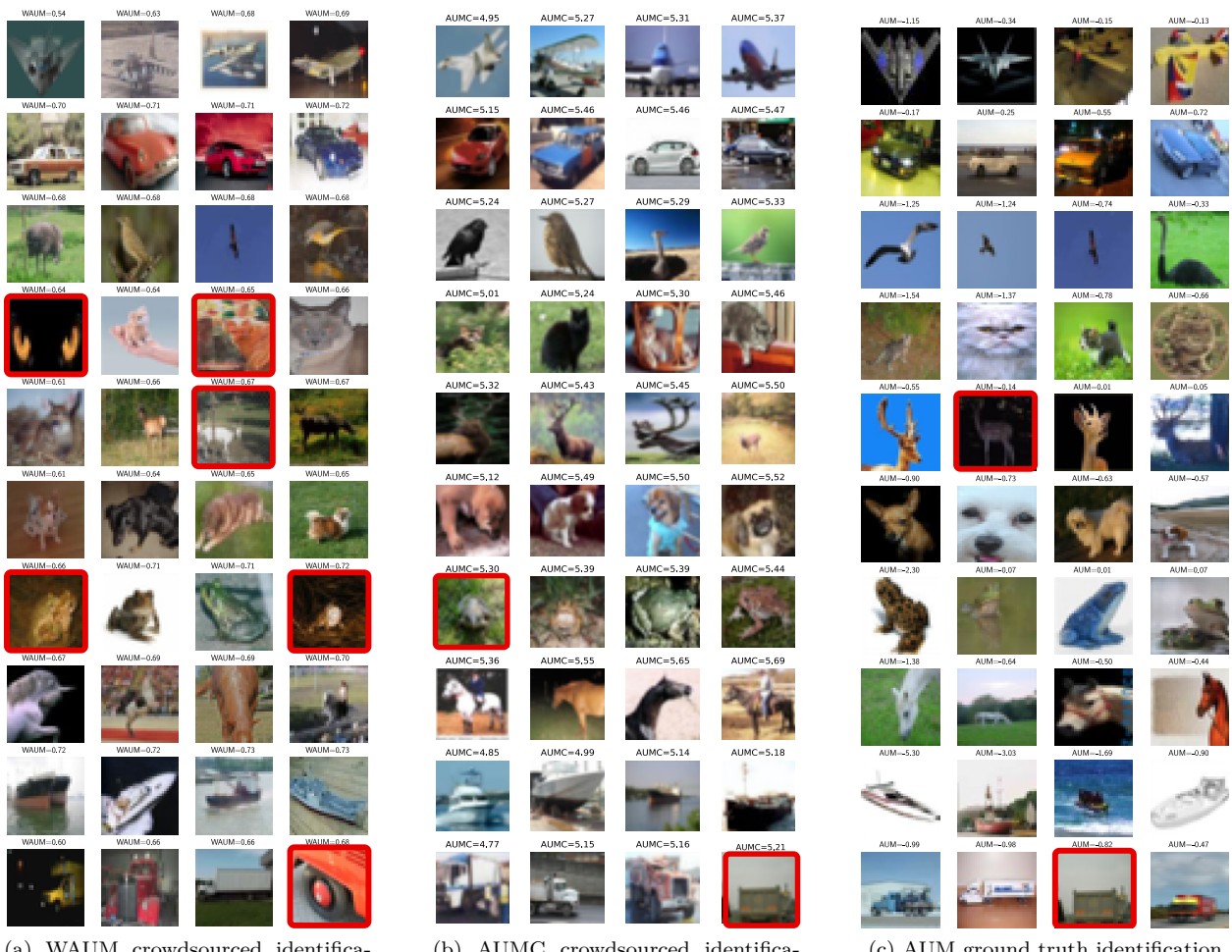

(a) WAUM crowdsourced identification

(b) AUMC crowdsourced identification

(c) AUM ground truth identification

Figure 16: Comparison of the worse images detected by the WAUM, AUMC and classical AUM preprocessing step. Identification was computed with a ResNet-18 for 50 epochs using the parameters described in Section 4. Each row represents the class given by the unobserved ground truth label from the `CIFAR-10` dataset. Only the AUM uses the ground truth label, other methods are based on the crowdsourced labels only. Images framed in red can be hard to classify.

#### D.2.2 The `LabelMe` dataset

Another real dataset in the crowdsourced image classification field that can be used is the `LabelMe` crowdsourced dataset created by Rodrigues & Pereira (2018). This dataset consists of $n_{\text{task}} = 1000$ training images

---

[7]Note that attention checks occurred every 20 trial for each worker, for tasks whose labels were known. They have been removed from the dataset since the corresponding images are not available.

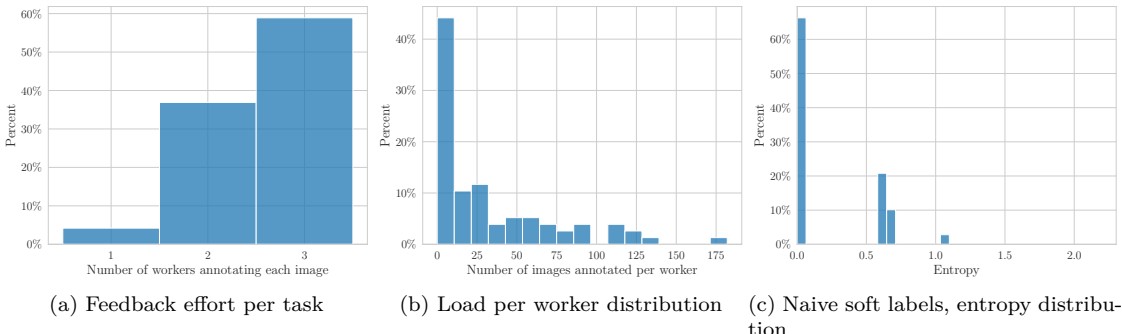

Figure 17: `LabelMe`: dataset visualization

dispatched among $K = 8$ classes: `highway`, `insidecity`, `tallbuilding`, `street`, `forest`, `coast`, `mountain` or `open country`. The validation set has 500 images and the test set has 1188 images. The whole training tasks have been labeled by $n_{\texttt{worker}} = 59$ workers, each task having between one and three given (crowd-sourced) labels. In particular, 42 tasks have been labeled only once, 369 tasks have been labeled twice and 589 received three labels. This is a way sparser labeling setting than the `CIFAR-10H` dataset.

Also, note that the `LabelMe` dataset has classes that overlap and thus lead to intrinsic ambiguities. This is the reason why the CoNAL strategy was introduced by Chu et al. (2021), see details in Appendix A.7. For example, the classes `highway`, `insidecity`, `street` and `tallbuilding` (in rows) are overlapping for some tasks: some cities have streets with tall buildings, leading to confusion as shown in Figure 10. The proposed feature aware aggregation using the WAUM leads to better performance in test accuracy and calibration as illustrated in Table 5.

### D.2.3 The `Music` dataset

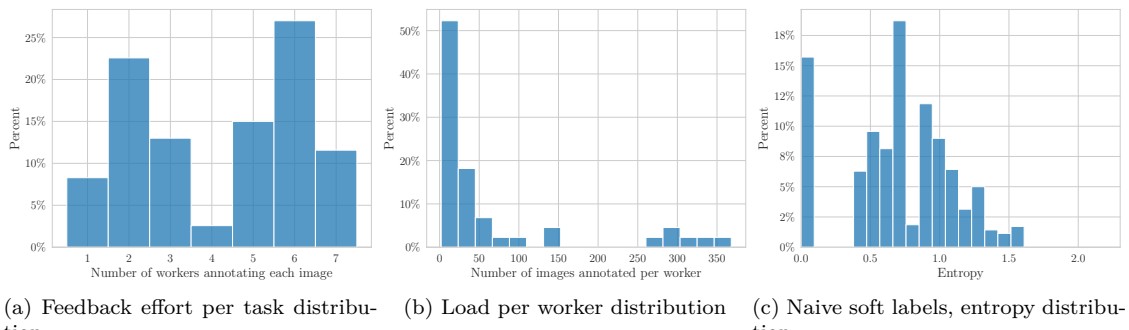

Figure 18: Music: dataset visualization

Rodrigues et al. (2014) released a crowdsourced dataset of audio files. The goal of this classification task was to decide the genre of 30 seconds musical excerpts. Number of tasks is $n_{\texttt{task}} = 700$. The $n_{\texttt{worker}} = 44$ workers had $K = 10$ possible labels: `blues`, `classical`, `country`, `disco`, `hiphop`, `jazz`, `metal`, `pop` and `reggae`. Each audio file was labeled by between 1 and 7 workers. To test the results, a dataset of 299 labeled clips is used (originally 300, but one file is known to be corrupted). Instead of working with the original audio files, we have used Mel spectrograms, openly available[8], to rely on standard neural networks architecture for image classification.

---

[8]https://www.kaggle.com/datasets/andradaolteanu/gtzan-dataset-music-genre-classification?datasetId=568973

| Strategy | $\text{Acc}_{\text{test}}(\%)$ | $1 - \text{ECE}$ |
|---|---|---|
| MV | $85.4 \pm 1.0$ | $\mathbf{0.864 \pm 0.01}$ |
| MV + AUMC | $86.0 \pm 1.1$ | $0.859 \pm 0.01$ |
| MV + WAUM | $\mathbf{86.1 \pm 0.9}$ | $0.858 \pm 0.02$ |
| NS | $86.1 \pm 1.0$ | $0.862 \pm 0.01$ |
| NS + AUMC | $87.2 \pm 0.8$ | $0.882 \pm 0.01$ |
| NS + WAUM | $\mathbf{88.1 \pm 1.0}$ | $\mathbf{0.890 \pm 0.02}$ |
| DS | $86.8 \pm 0.5$ | $\mathbf{0.877 \pm 0.01}$ |
| DS + AUMC | $86.3 \pm 0.5$ | $0.841 \pm 0.02$ |
| DS + WAUM | $\mathbf{87.2 \pm 0.6}$ | $0.862 \pm 0.02$ |
| GLAD | $87.1 \pm 0.9$ | $0.881 \pm 0.01$ |
| GLAD + AUMC | $87.6 \pm 1.1$ | $0.861 \pm 0.03$ |
| GLAD + WAUM | $\mathbf{88.2 \pm 0.8}$ | $\mathbf{0.885 \pm 0.02}$ |
| WDS | $85.6 \pm 0.7$ | $0.838 \pm 0.02$ |
| WDS + AUMC | $86.7 \pm 0.7$ | $0.862 \pm 0.02$ |
| WDS + WAUM | $\mathbf{87.1 \pm 0.8}$ | $\mathbf{0.871 \pm 0.01}$ |
| CrowdLayer | $85.4 \pm 4.2$ | $0.858 \pm 0.04$ |
| CrowdLayer + AUMC | $87.1 \pm 3.5$ | $0.809 \pm 0.05$ |
| CrowdLayer + WAUM | $\mathbf{87.5 \pm 3.2}$ | $\mathbf{0.860 \pm 0.03}$ |
| CoNAL$(\lambda = 0)$ | $88.1 \pm 1.0$ | $0.881 \pm 0.01$ |
| CoNAL$(0)$ + AUMC | $89.1 \pm 1.1$ | $\mathbf{0.903 \pm 0.02}$ |
| CoNAL$(0)$ + WAUM | $\mathbf{89.2 \pm 1.0}$ | $0.892 \pm 0.01$ |
| CoNAL$(\lambda = 10^{-4})$ | $86.2 \pm 6.4$ | $0.865 \pm 0.06$ |
| CoNAL$(10^{-4})$ + AUMC | $88.4 \pm 2.3$ | $0.884 \pm 0.04$ |
| CoNAL$(10^{-4})$ + WAUM | $\mathbf{90.0 \pm 0.8}$ | $\mathbf{0.901 \pm 0.01}$ |

Table 5: Ablation study on `LabelMe` using the VGG backbone: $\alpha = 0.01$. Errors represented are standard deviations.

| Strategy | $\text{Acc}_{\text{test}}(\%)$ | $1 - \text{ECE}$ |
|---|---|---|
| MV | $59.9 \pm 1.23$ | $0.631 \pm 0.01$ |
| MV + AUMC | $\mathbf{62.0 \pm 1.23}$ | $\mathbf{0.650 \pm 0.02}$ |
| MV + WAUM | $61.1 \pm 2.35$ | $0.624 \pm 0.02$ |
| NS | $59.9 \pm 1.40$ | $0.624 \pm 0.02$ |
| NS + AUMC | $59.9 \pm 1.41$ | $0.640 \pm 0.02$ |
| NS + WAUM | $\mathbf{62.1 \pm 2.18}$ | $\mathbf{0.642 \pm 0.01}$ |
| DS | $\mathbf{62.9 \pm 1.72}$ | $\mathbf{0.661 \pm 0.01}$ |
| DS + AUMC | $61.5 \pm 2.22$ | $0.659 \pm 0.01$ |
| DS + WAUM | $62.1 \pm 2.81$ | $0.640 \pm 0.02$ |
| GLAD | $61.5 \pm 1.72$ | $0.639 \pm 0.01$ |
| GLAD + AUMC | $\mathbf{61.6 \pm 0.93}$ | $\mathbf{0.664 \pm 0.01}$ |
| GLAD + WAUM | $61.5 \pm 1.23$ | $0.645 \pm 0.01$ |
| WDS | $60.2 \pm 1.66$ | $0.652 \pm 0.01$ |
| WDS + AUMC | $62.9 \pm 2.67$ | $0.647 \pm 0.03$ |
| WDS + WAUM | $\mathbf{63.1 \pm 0.91}$ | $\mathbf{0.660 \pm 0.02}$ |
| CrowdLayer | $63.2 \pm 1.34$ | $0.615 \pm 0.02$ |
| CrowdLayer + AUMC | $\mathbf{63.3 \pm 2.54}$ | $0.617 \pm 0.04$ |
| CrowdLayer + WAUM | $63.2 \pm 2.46$ | $\mathbf{0.680 \pm 0.03}$ |
| CoNAL$(\lambda = 0)$ | $64.2 \pm 0.91$ | $0.660 \pm 0.02$ |
| CoNAL$(0)$ + AUMC | $64.3 \pm 0.88$ | $\mathbf{0.735 \pm 0.01}$ |
| CoNAL$(0)$ + WAUM | $\mathbf{64.5 \pm 0.76}$ | $\mathbf{0.735 \pm 0.01}$ |
| CoNAL$(\lambda = 10^{-4})$ | $64.2 \pm 0.55$ | $0.639 \pm 0.06$ |
| CoNAL$(10^{-4})$ + AUMC | $64.1 \pm 0.74$ | $\mathbf{0.745 \pm 0.02}$ |
| CoNAL$(10^{-4})$ + WAUM | $\mathbf{64.4 \pm 0.78}$ | $0.726 \pm 0.02$ |

Table 6: Ablation study on `Music` using the VGG backbone: $\alpha = 0.05$. Errors represented are standard deviations.

# E Algorithmic details on the neural network training

Experiments can be reproduced using the code available at `https://github.com/peerannot/peerannot` from the `peerannot` library, which is briefly described below:

- The `identification` module is used to explore datasets tasks and workers. Tasks can be explored thanks to the entropy of the label distribution, the WAUM or the AUMC. Workers can be evaluated thanks to the Spam-score of Raykar & Yu (2011), the trace of the DS estimated matrices, GLAD's parameters among other.

- The `aggregate` module is used to produce aggregated labels from multiple answered labels. The labels can then be used for training a neural network architecture from `Pytorch` using the `train` module.

- The `aggregate-deep` module is used for the CoNAL and CrowdLayer strategies. A neural network is directly learning from the crowdsourced tasks and labels without the aggregation step.

- Multiple datasets are ready to use, including `CIFAR-10H`, `LabelMe` and `Music`.

The documentation of the library is at `https://peerannot.github.io/`.

# F Results on LabelMe and Music datasets

In this section, we provide the raw results tables of the forest plots (Figure 8 and Figure 11) the real datasets in Table 5 on the `LabelMe` and in Table 6 for the `Music` datasets. The errors presented are standard deviations.

## G  Margin comparison

In the AUM, AUMC and WAUM formulae, we consider a margin from Yang & Koyejo (2020) (denoted $\psi_5$ in the original article) that has better theoretical properties for top-$k$ classification but that is not the margin proposed in Pleiss et al. (2020) ($\psi_1$). Indeed, our margin in the AUM is written as:

$$\sigma_{y_i}^{(t)}(x_i) - \sigma_{[2]}^{(t)}(x_i) \ ,$$

instead of

$$\sigma_{y_i}^{(t)}(x_i) - \max_{k \neq y_i} \sigma_k^{(t)}(x_i) \ .$$

Using the `CIFAR-10H` dataset, we can compare the identified tasks using each margin. Note that in the library used (and briefly described in Appendix E) switching from the original margin to the top-$k$ based margin is executed with the argument `use_pleiss=True` or `use_pleiss=False` with the WAUM, AUM and AUMC. A comparison of the images with lowest AUM is provided in Figure 21. Similar visual comparison on the `CIFAR-10H` dataset is provided in Figure 24.

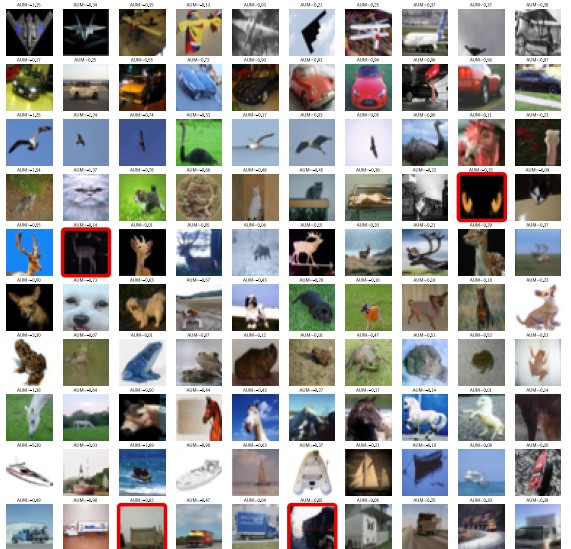 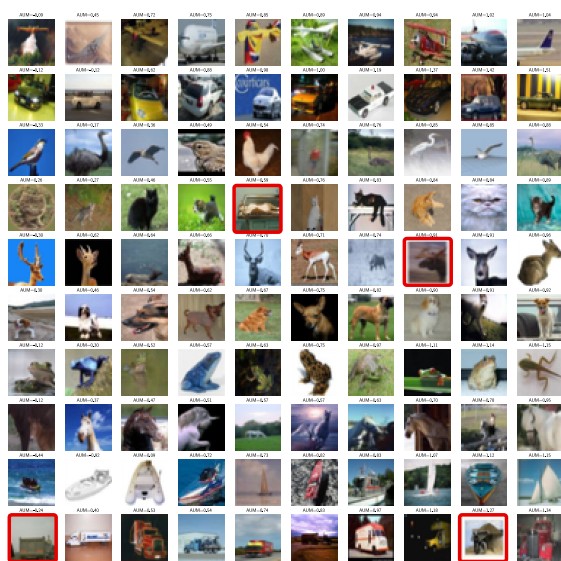

Figure 19: Lowest AUM with $\psi_1$ margin            Figure 20: Lowest AUM with $\psi_5$ margin

Figure 21: Comparison of the images with lowest AUM in `CIFAR-10H` dataset using the margin from Pleiss et al. (2020) ($\psi_1$) or the margin for top-1 classification from Yang & Koyejo (2020) ($\psi_5$). Both margins yield similar results.

Furthermore, we provide an ablation study on top-2 accuracy scores using the WAUM with $\psi_1$ or $\psi_5$ margin on the `LabelMe` dataset in Table 7. We use the crossentropy loss during the training phase. With $\psi_5$, the top-2 margin writes $\sigma_{y_i}^{(t)}(x_i) - \sigma_{[3]}^{(t)}(x_i)$ as indicated in Section 3.1. We compute the top-2 accuracy *i.e.* the accuracy in recovering the true label as the first or second predicted label by the classifier. However, note that this dataset has $K = 8$ classes, hence we do not report top-5 accuracy as all strategies perform similarly. We notice that performance on this dataset are similar for most strategies between the two margins used for pruning.

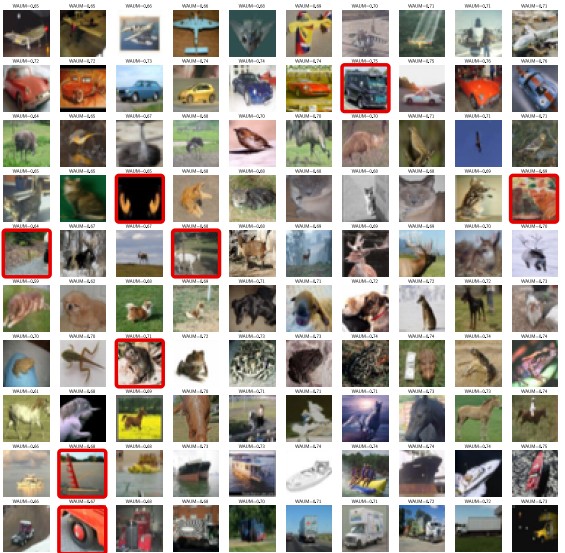

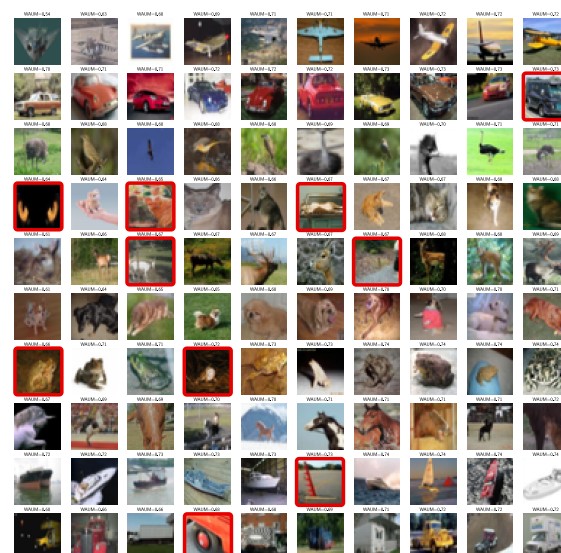

Figure 22: Lowest WAUM with $\psi_1$

Figure 23: Lowest WAUM with $\psi_5$

Figure 24: Comparison of the images with lowest WAUM in `CIFAR-10H` dataset using the margin from Pleiss et al. (2020) ($\psi_1$) or the margin for top-1 classification from Yang & Koyejo (2020) ($\psi_5$). Both margins also lead to similar results.

Table 7: Top-2 Accuracy comparison on the `LabelMe` dataset using the modified VGG-16 backbone and the same hyperparameters as in Section 4. Results are averaged over 5 repetitions, errors are standard deviations.

| Strategy | Top-2 no pruning | Top-2 WAUM($\psi_1$) | Top-2 WAUM($\psi_5$) |
|---|---|---|---|
| MV | $91.25 \pm 2.01$ | $91.17 \pm 2.12$ | $92.02 \pm 2.08$ |
| NS | $90.92 \pm 1.53$ | $90.41 \pm 2.77$ | $89.91 \pm 1.08$ |
| DS | $89.98 \pm 1.12$ | $90.24 \pm 0.92$ | $91.41 \pm 0.99$ |
| GLAD | $90.78 \pm 0.98$ | $91.34 \pm 1.59$ | $90.49 \pm 0.38$ |
| WDS | $89.56 \pm 1.76$ | $90.82 \pm 1.81$ | $91.16 \pm 2.78$ |
| CrowdLayer | $87.45 \pm 2.03$ | $88.33 \pm 1.49$ | $88.57 \pm 2.51$ |
| CoNAL($\lambda = 0$) | $92.34 \pm 0.74$ | $89.49 \pm 0.53$ | $94.30 \pm 1.32$ |
| CoNAL($\lambda = 10^{-4}$) | $91.68 \pm 1.01$ | $94.10 \pm 0.9$ | $94.93 \pm 0.76$ |

