# OpenReview forum: "Identify Ambiguous Tasks Combining Crowdsourced Labels by Weighting Areas Under the Margin"
_TMLR — Accepted by TMLR_

### Review · Reviewer_p4kV · 2023-12-07

**Summary Of Contributions:**

This paper considers the problem of classification with crowdsourced labels. Each annotation task is to label an example in the dataset, and some tasks may have noisier labels than others. The authors propose to leverage the Area Under the Margin (AUM) ranking to filter out tasks with high ambiguity, in order to improve generalization on the test set. Experiments show that two variants (hard label AUMC and soft label WAUM) are compatible with standard crowdsourced data aggregation methods, leading to improved classification accuracy and calibration error on synthetic and standard image classification datasets.

**Audience:**

Yes

**Broader Impact Concerns:**

Concerns are addressed sufficiently at the end of the submission.

**Claims And Evidence:**

Yes

**Requested Changes:**

### Critical
- Justify the use of non-standard margin in Equation (2)
- Report top-$k$ accuracy


### Non-critical
- Ablation with standard margin in place of Equation (2).
- Include AUMC method in Table 1
- Comment on whether methods in Section 4 are common/simple baselines or state-of-the-art methods for crowdsourced label aggregation
- Cite and discuss/compare with recent work which combines data pruning and crowdsourcing, Xing et al. (2023).

Xing et al. [Training with Low-Label-Quality Data: Rank Pruning and Multi-Review](https://dmlr.ai/assets/accepted-papers/7/CameraReady/Improving_Scammer_Detection__Algorithm_and_Multiple_Reviews%20(3).pdf), in ICML Workshop on Data-centric Machine Learning Research, 2023.

**Strengths And Weaknesses:**

### Strengths
- Straightforward combination of existing work
- Well-organized, with clear notation
- Thorough experiments show clear improvements when combined with multiple aggregation baselines

### Weaknesses

- It is unclear why the authors define AUM with $\psi_5$ in Table 1 of Yang & Koyejo (2020) rather than with the much more standard margin definition $\psi_1$. The argument in Section 3.2 about top-$k$ settings is not convincing because the main metric considered is top-1 accuracy. The authors should justify this, report top-$k$ accuracy, and consider performing an ablation with $\psi_1$ margin.

- Typo: "which is weights the margins" should be "which weights the margins"


F. Yang and S. Koyejo. On the consistency of top-k surrogate losses. In ICML, pp. 10727–10735, 2020.

---

### Review · Reviewer_GkhP · 2023-12-08

**Summary Of Contributions:**

The authors propose a method for identifying ambiguous tasks in datasets where each task has been performed by multiple crowdworkers. The proposed method, Weighted Areas Under the Margin (WAUM), is an adapted version of Area Under the Margin (AUM), which averages AUMs weighted by task-dependent scores. They perform experiments on both synthetic and real datasets showing that the proposed method is able to identify ambiguous tasks and subsequently improve the performance of classifiers by training them on cleaned versions of the underlying dataset (i.e., with the ambiguous tasks removed).

**Audience:**

Yes

**Broader Impact Concerns:**

I found the Broader Impact Statement included in the paper to be sufficient.

**Claims And Evidence:**

Yes

**Requested Changes:**

1. Algorithm 1 also requires training a classifier as part of step 4 right (i.e., obtaining the trust scores). If that's the case, then you should clarify that in the algorithm and also in the "dataset pruning" paragraph. Relatedly, this approach to computing trust scores assumes that the classifier that you train is good. How good does it need to be? You should elaborate a bit more on this in Section 3.3 in my opinion.

2. In the "dataset pruning" section you mention that you are pruning based on a quantile of the WAUM scores. Why did you choose to use a quantile instead of a threshold on the values themselves? I am a bit concerned that using a quantile means you will always throw away some data even if it is good. I would at least clarify that in the paper and also provide some justification for why you are using a quantile instead of a threshold on the WAUM value itself.

3. In the implementation details section of your experiments, you may want to clarify that by "three dense layers' artificial network" what you actually mean a three-layer perceptron (i.e., a three-layer MLP). There is also a typo in the penultimate sentence of that paragraph ("wether" -> "whether").

4. In the experiments section when you present tables with results you have some error bars in "+-" values. You should say what those quantities are (i.e., are they standard deviations, standard errors, etc.), ideally in the caption of each table. In those same captions you should also explain what the bolded values represent, and especially so for tables 2, 3, and 4 where they are the largest value per section. I would also recommend you separate those sections in those tables with lines/borders to make it even more clear.

5. Related to those same tables, a lot of the results do not seem statistically significant. I would use a visual cue for representing those that are statistically significant (e.g., bold and red or something to differentiate it from just bold).

6. In Section 3.2 you say "... the higher the AUM, the more confident the prediction is in the assigned label. Hence, the lower the AUM, the more likely the label is wrong ..." This is confusing "confidence" with "correctness" which is a strong assumption. I think you should not make that statement.

7. In the introduction, you sometimes use the word "harmful" to characterize the instances of the data that you prune. I think this can be misinterpreted and a more accurate characterization would be to use "ambiguous" throughout.

8. There is some related work that you might find interesting and pertinent to this work, starting with Platanios, et al. "Estimating Accuracy from Unlabeled Data" (2014), that is followed by 2-3 more papers by the same authors exploring variations of this problem.

**Strengths And Weaknesses:**

### Strengths

- The paper tackles an interesting problem.
- The paper is well-written and easy to follow.
- The proposed method makes sense intuitively.

### Weaknesses

- The experiment results do not seem statistically strong, though I'm not sure if I'm assessing this correctly based on the information provided in the paper.
- The paper is lacking comparison to simple baselines for pruning the underlying dataset. A very simple baseline could be to prune based on the variance of the crowdworker labels for each task.
- The assumptions that the proposed method makes are not discussed.

---

### Review · Reviewer_r8Ez · 2024-02-27

**Summary Of Contributions:**

A novel extension of the Area Under the Margin (AUM) method for crowdsourcing data with multiple labels called Weighted Areas Under the Margin (WAUM) is proposed by using task-dependent scores. It can be used to identify harmful data points in the dataset, so as a step 2, the user can prune the dataset based on the WAUM score of each data point. Also a naive baseline called AUMC is proposed. Experiments show the behavior of the proposed methods. Additionally, experiments show comparison with many baseline methods.

**Audience:**

Yes

**Broader Impact Concerns:**

I have no societal concerns with respect to this paper.

**Claims And Evidence:**

Yes

**Requested Changes:**

I wrote some comments in the "weaknesses" in the previous section. I believe it would be beneficial for the paper to take these two points into consideration.

A very minor comment: In the rebuttal letter, the authors wrote they updated the paper to avoid using the term "harmful". It seems that the version I am reading still uses this term 3 times.

**Strengths And Weaknesses:**

Strengths
- A novel method is proposed that empirically works better than baselines (and also another proposed naive baseline.)
- Experiments are conducted for both synthetic ones and benchmark datasets. Synthetic datasets provide insights into the behavior of the proposed method. Benchmark datasets include a scenario with higher number of labels per data (CIFAR-10H) and fewer number of labels per data (LabelMe, Music).
- Well-documented code is provided in the submission.
- It is interesting that the method is applicable not only for label noise and out-of-distribution scenarios, but also for class-overlap type of situations (shown with the experiments with LabelMe dataset.)

Weaknesses
- I may have missed it, but the details about the experimental setup were not clear. For example, what is the test dataset in CIFAR-10H? (I understand the training and validation dataset which is explained in page 10)
- A related comment is that, it would be more appropriate if we can check the accuracy with the test dataset, for the best hyper-parameter alpha chosen based on the validation dataset. (For example in Section 4.1, it seems there is only train and test dataset (but no validation dataset.) This may be helpful to avoid overfitting on the evaluation dataset.

---

> ### Author Response · Authors · 2024-03-06
> **Response to reviewer r8Ez**
>
> We thank you for the valuable feedback on this paper and the pointed strengths.
> As for your concerns:
>
> - **Experimental setup**: For the train/validation/test sets, we moved more details in Section 4.2, they were previously in Appendix D.2  on the real datasets extensive description. To answer shortly: both CIFAR-10H train and validation sets are composed of the images from the original CIFAR-10 test set; CIFAR-10H test set is the train set of CIFAR-10 (the creators switched the sets to have less tasks to crowdsource and reduce their costs).
> - **Validation set and $\\alpha$ calibration**: We agree that one should not chose the $\\alpha$ hyperparameter based on the test set at the risk of overfitting. We did not perform the tuning that way, but we reckon that some details were missing. We now elaborate on this Section 3.3 in the paragraph "Dataset Pruning" about the impact of each parameter. We added more details and renamed this paragraph to insist on hyperparameter calibration using the validation set. For real datasets, we used the grid described in Section 3.3 with $\alpha\\in\\{0.1, 0.05, 0.01\\}$ (smalle $\alpha$'s would almost not remove any task and larger one lead to remove too many) and reported the test values for the best performance on the validation set ($\\alpha=0.01$ for LabelMe, CIFAR-10H and $\\alpha=0.05$ on Music). Note that in general, dataset annotated by humans have roughly between $1$ and $5\%$ of errors (Northcutt et. al, 2021) and the choice of $\alpha$ should reflect that. We did not report each validation accuracy, as this would add three more numerical results per strategy and reduces readability.
> For example on LabelMe, for the WDS aggregation strategy we have (averaged over $5$ repetitions):
>
> | Strategy                | Validation accuracy | Test accuracy |
> | ----------------------- | ------------------- | ------------- |
> | WDS + WAUM(alpha=0.005) | 84.3                | 85.9          |
> | WDS + WAUM(alpha=0.01)  | 85.2                | 87.1          |
> | WDS + WAUM(alpha=0.05)  | 83.6                | 86.7          |
> | WDS + WAUM(alpha=0.1)   | 82.4                | 85.3          |
>
>
> Note that, indeed in Section 4.1 (on simulations) we did not use a validation set. These numerical experiments were toys examples to illustrate the impact of the pruning with different strategies and hyperpameters. For pedagogical insights, we chose to show in Fig. 5 how the choice of $\alpha$ impacts the pruning on the training set, and the performance is not what was targeting in this experiments. For the three real datasets in Section 4.2 though, we have used the available validation set.
>
> Concerning the use of the "harmful" term, thank you for pointing this out. Only one of the three was about the data pruning. To completely remove this word and the associated concerns from our work, we rephrased these sentences.
>
> Thank you again and we hope these clarifications will help.

---

### Decision · Action_Editor_mnU7 · 2024-04-11

**Recommendation:** Accept as is

**Comment:**

The reviewers were satisfied with the updates of the paper made during the review phase and all recommended acceptance of the paper. Hence, as the paper also tackles a relevant and interesting problem, I am recommending acceptance in line with the reviewers' recommendations.

**Audience:**

Yes, the paper is relevant to a subset of TMLR's audience.

**Claims And Evidence:**

The main claim is that the proposed  Weighted Areas Under the Marg (WAUM) can help to identify and discard ambiguous tasks and, thereby, lead to better generalization performance. This claim is supported by several experimental results on synthetic and real-world data (not all differences are statistically significant though).

---

> ### Author Response · Authors · 2024-04-15
> **Thank you**
>
> We wanted to thank all three reviewers and the Action Editor for their reviews that improved the now accepted paper and for their time.
> We updated the submission with the camera-ready version and the codes are available in supplementary materials (and online via the peerannot library).
>
> Thank you again
> The authors